# Durvalumab after Sequential High Dose Chemoradiotherapy versus Standard of Care (SoC) for Stage III NSCLC: A Bi-Centric Trospective Comparison Focusing on Pulmonary Toxicity

**DOI:** 10.3390/cancers14133226

**Published:** 2022-06-30

**Authors:** Romana Wass, Maximilian Hochmair, Bernhard Kaiser, Brane Grambozov, Petra Feurstein, Gertraud Weiß, Raphaela Moosbrugger, Felix Sedlmayer, Bernd Lamprecht, Michael Studnicka, Franz Zehentmayr

**Affiliations:** 1Department of Pulmonology, Paracelsus Medical University, A-5020 Salzburg, Austria; romana.wass@kepleruniklinikum.at (R.W.); g.weiss@salk.at (G.W.); r.moosbrugger@salk.at (R.M.); m.studnicka@salk.at (M.S.); 2Department of Pulmonology, Kepler University Hospital, A-4020 Linz, Austria; bernhardkaiser@gmx.at (B.K.); bernd.lamprecht@kepleruniklinikum.at (B.L.); 3Department of Respiratory and Critical Care Medicine, Karl Landsteiner Institute of Lung Cancer Research and Pulmonary Oncology, Klinik Floridsdorf, A-1210 Vienna, Austria; maximilian.hochmair@gesundheitsverbund.at; 4Department of Radiation Oncology, Paracelsus Medical University, A-5020 Salzburg, Austria; b.grambozov@salk.at (B.G.); f.sedlmayer@salk.at (F.S.); 5Department of Radiation Oncology, Klinik Ottakring, A-1160 Vienna, Austria; petra.feurstein@gesundheitsverbund.at; 6radART—Institute for Research and Development on Advanced Radiation Technologies, Paracelsus Medical University, A-5020 Salzburg, Austria

**Keywords:** high dose radiation, durvalumab, toxicity, pneumonitis, chemoradiotherapy

## Abstract

**Simple Summary:**

The standard of care (SoC) for patients with unresectable stage III non-small-cell lung cancer is concurrent chemoradiation followed by durvalumab. Because of co-morbidities, however, the concurrent approach is only amenable for about one-third of patients. While sequential regimens are usually regarded as palliative, these schedules applied in a dose-escalated mode may be part of a curative approach. As the combination of high-dose radiation and durvalumab remains a question of ongoing research, this retrospective study aims at evaluating pulmonary side effects after sequential chemoradiotherapy with high-dose irradiation compared to SoC. Radiation dose escalation showed no excess pulmonary toxicity such as pneumonitis but tendentially better intrathoracic control, suggesting that this alternative approach is safe and feasible.

**Abstract:**

Introduction: The standard of care (SoC) for unresectable stage III non-small-cell lung cancer (NSCLC) is durvalumab maintenance therapy after concurrent chemoradiation in patients with PD-L1 > 1%. However, the concurrent approach is only amenable for about one-third of patients due to co-morbidities. Although sequential regimens are usually not regarded as curative, these schedules applied in a dose-escalated manner may be similarly radical as SoC. As combining high-dose radiation and durvalumab remains a question of debate this retrospective bi-center study aims to evaluate pulmonary toxicity after high-dose chemoradiotherapy beyond 70 Gy compared to SoC. Patients and Methods: Patients with NSCLC stage III received durvalumab after either sequential high-dose chemoradiation or concomitant SoC. Chemotherapy consisted of platinum combined with either pemetrexed, taxotere, vinorelbine, or gemcitabine. The primary endpoint was short-term pulmonary toxicity occurring within six months after the end of radiotherapy (RT). Results: A total of 78 patients were eligible for this analysis. 18F-FDG-PET-CT, cranial MRT, and histological/cytological verification were mandatory in the diagnostic work-up. The high-dose and SoC group included 42/78 (53.8%) and 36/78 (46.2%) patients, respectively, which were matched according to baseline clinical variables. While the interval between the end of RT and the start of durvalumab was equal in both groups (*p* = 0.841), more courses were administered in the high-dose cohort (*p* = 0.031). Pulmonary toxicity was similar in both groups (*p* = 0.599), whereas intrathoracic disease control was better in the high-dose group (local control *p* = 0.081, regional control *p* = 0.184). Conclusion: The data of this hypothesis-generating study suggest that sequential high-dose chemoradiation followed by durvalumab might be similar to SoC in terms of pulmonary toxicity and potentially more effective with respect to intra-thoracic disease control. Larger trials with a prospective design are warranted to validate these results.

## 1. Introduction

Lung cancer is one of the most frequent malignancies and the leading cause of cancer death worldwide [1,2]. About 80 percent of patients are affected by tumors with non-small cell histologies (NSCLC). Approximately one-third present with locally advanced UICC stage III disease at the time of diagnosis, which comprises a heterogeneous group of patients with 5-year survival rates ranging between 5 and 43% [3,4].

Until 2017, the standard of care (SoC) for patients with a good performance status was platinum-based doublet chemotherapy administered concurrently with radiotherapy (RT) given in 2 Gy fraction to a total dose of 60 Gy. While local control rates of 70% can be achieved with this therapeutic approach, progression-free (PFS) and overall survival (OS) are still poor [5]. However, for toxicity reasons, this treatment strategy is only amenable for about 30% of cases [6]. Elderly patients or those in reduced general condition are recommended to receive sequential regimens, which are usually not regarded as curative unless irradiation is administered in a dose-escalated manner (reviewed by [7,8]).

Based on the results of the PACIFIC trial [1,9], which demonstrated the superiority of durvalumab, a selective high-affinity human IgG1 monoclonal antibody that blocks programmed death-ligand 1 (PD-L1) binding to programmed death 1 (PD-1) and CD80 over placebo, patients without disease progression after chemoradiotherapy (CRT) should receive durvalumab maintenance therapy for one year. Immune checkpoint inhibition (ICI) significantly prolonged PFS (median 18.3 months; HR 0.44) and OS (median not reached, HR 0.54) [1]. The absolute increase in the 2-year OS was 19.2% [9]. The follow-up publications of the 4- and 5-year data corroborate these encouraging results [4,10]. Apart from the PACIFIC trial and its follow-up analyses, real-world data (RWD) have been published in the past two years [11], whose results are in line with the randomized data.

ICI consolidation has become the SoC [12,13], however, altering the immune response in this way, may come at the cost of immune-related adverse events (irAEs), possibly affecting multiple organs. Checkpoint inhibitor pneumonitis (CIP) is a rare but clinically highly relevant irAE that can significantly impair quality of life and can be potentially life-threatening. Overall, the development of irAEs is associated with improved ICI efficacy and OS [14,15], yet severe CIP decreased PFS and OS in NSCLC patients treated with anti-PD(L)1 therapy [16]. The incidence of CIP varies between 2–5% in clinical trials and meta-analyses but can be as high as 13–19%, according to RWD [17,18,19]. ECOG > 2 [16], pre-existing interstitial lung disease (ILD) [16,20,21,22], squamous cell histology [20], combination therapies of ICI [20] and EGFR TKIs [23,24] are discussed as potential risk factors for CIP. Additionally, prior thoracic RT may also be a reason for an increased rate of pulmonary side effects in patients with ICI [20]. In fact, the PACIFIC trial demonstrated that durvalumab increased the incidence of any grade pulmonary toxicity by roughly 50% compared to placebo (46.1% vs. 31.2%) [1,9].

Although the PFS and OS data from the PACIFIC trial present a new milestone in the treatment of stage III NSCLC, the intrathoracic failure rate of 36.6% remains high [25], which may be due to the relatively low total radiation doses of 54 to 66 Gy. Landman et al. were the first to successfully address this issue by combining radiation dose-escalation and durvalumab [26]. Although the cohort from this single-center experience was small, the loco-regional relapse rate in these 39 patients could be reduced to 21%. These results imply that dose-escalation in the context of durvalumab consolidation may increase intrathoracic disease control without excess toxicity [26].

Based on these results [26], we hypothesized that durvalumab following sequential high-dose RT can be applied without excess pulmonary toxicity compared to SoC. Therefore, the aim of this retrospective analysis was to evaluate pulmonary toxicity in patients treated with durvalumab maintenance after sequential high-dose radiochemotherapy compared to SoC concomitant CRT.

## 2. Patients and Methods

### 2.1. Patients

Patients with histologically or cytologically proven stage III NSCLC (8th edition of the TNM staging system) were eligible for this analysis, which was approved by the ethics committee of the Federal State of Salzburg (approval number: 1113/2021). The diagnostic work-up for each patient comprised ^18^F-FDG-PET-CT, cranial MRT, bronchoscopy, and endobronchial ultrasound-guided transbronchial needle aspiration (EBUS-TBNA) in the mediastinum and discussion in the interdisciplinary tumor board with pulmonologists, medical oncologists, radiologists, thoracic surgeons, and radiation-oncologists. Pulmonary function was assessed before radiation treatment by body plethysmography, blood gas analysis, and DLCO. Patients had to have a minimum FEV1 of 0.8 L before treatment. In the high-dose CRT group, pulmonary function tests (PFTs) were repeated at the end of RT and six months after completion of RT. High-dose and SoC patients were matched according to the following criteria: age (+/−10 years), sex, ECOG, smoking status, and histology. This approach is similar to a previous study by Johnson [27].

### 2.2. Chemoradiotherapy

The analysis was designed as a two-armed bi-centric cohort study comparing a high-dose CRT group (Salzburg) with patients who received SoC, i.e., concomitant CRT (Vienna). The aim of this investigation was to assess and compare pulmonary toxicity in both groups.

Patients at the Salzburg center were traditionally treated with sequential high-dose RT after induction chemotherapy [28,29,30]. The patients included in the current analysis received volumetric arc therapy (VMAT). Prior to radiation therapy, patients usually received two cycles of platinum-based chemotherapy containing cisplatin (75 mg/m^2^) or carboplatin AUC 5 and pemetrexed (500 mg/m^2^) or gemcitabine (1000 mg/m^2^). Irradiation treatment was either administered in twice-daily fractions of 1.8 Gy to a total dose of 73.8 Gy [30] or in 3 Gy fractions to a total dose of 66 Gy. For purposes of comparability with standard regimens, the biologically equivalent dose in 2 Gy fractions (*EQD*_2_) was calculated according to the following formalism with *D* as the total physical dose, *d* as a single fraction dose, and *α*/*β* as a tissue-dependent parameter.
$$EQD_{2}=D\times \frac{d+\alpha /\beta }{2+\alpha /\beta }$$

Assuming an *α*/*β*-value of 10 Gy for tumor tissue, the high-dose regimens amount to an *EQD*_2_ of 72.6 Gy and 71.5 Gy, respectively, which is in the range of the dose-escalation arm of the RTOG 0617 study with 74 Gy in 2 Gy fractions [31]. For gross tumor volume (GTV) and clinical target volume (CTV) delineation, the pre-therapeutic staging ^18^F-FDG-PET-CT scan was co-registered with the planning CT scan. The planning target volume (PTV) included the CTV with a 7 mm margin [30].

Patients treated in the Vienna cohort received concomitant CRT with 60 Gy in 2 Gy fractions (*EQD*_2_) combined with simultaneous systemic treatment. With the exception of Vinorelbin (25 mg/m^2^), the medicinal agents were identical to the Salzburg cohort. Likewise, the radiation treatment planning procedure was similar to the high-dose group, with the exception that the margins from CTV to PTV were 1.5–2 cm resulting in larger radiation volumes.

### 2.3. Immunotherapy

Based on the results of the PACIFIC trial [1,9] and international guidelines [12,13], bi-weekly durvalumab maintenance therapy (10 mg/kg) was initiated after CRT if the patient had no evidence of disease progression. This treatment was planned to be carried on for one year and discontinued only in case of disease progression, the start of alternative anticancer therapy, or unacceptable toxicity.

### 2.4. Follow-Up

After completion of immunotherapy maintenance, patients were included in the follow-up program. Contrast-CT scans of the chest and upper abdomen were performed every three months within the first two years and every six months thereafter. As an additional quality control for patients treated in the high-dose radiochemotherapy group, PFTs were performed at each follow-up visit in order to detect potential decreases in lung function that might be clinically irrelevant and therefore overlooked. If there were any signs of tumor progression, an additional ^18^F-FDG-PET-CT scan was performed.

### 2.5. Statistics and Endpoints

The current bi-center analysis was designed as a non-inferiority study. This type of analysis requires the definition of a cut-off value below which the experimental arm can be regarded as non-inferior. In the current study, this value was set at +20%, meaning that if the high-dose group had a maximum of 20% excess pulmonary toxicity compared to SoC (i.e., the confidence interval must not exceed 20%), it could be accepted as non-inferior (Appendix A). This cut-off value was based on the toxicity data of the original PACIFIC study [1]. Any grade pulmonary toxicity, which comprises pneumonitis—either related to radiation or immunotherapy—and pneumonia, was approximately 45% higher in the durvalumab arm than in the placebo group (47% vs. 32.5%). Based on these data, we allowed an excess of 20%, which is less than half of this value, to show non-inferiority. Additionally. we performed a power calculation based on the following assumptions: α-error 5%, power (1-β) 80%, 47% probability for pulmonary side effects in the control group [1] and 25% in the high-dose group [30,32]. The non-inferiority limit was set at 20%, as described above (www.sealedenvelope.com (accessed on 20 June 2020).

The primary endpoint was pulmonary toxicity, including pneumonitis and pneumonia within six months after completion of RT. Side effects were scored according to the Common Terminology Criteria for Adverse Events version (CTCAE) 5.0. Immune-related toxicities (irAEs) such as endocrinopathies, hepatitis, dermatitis, and colitis were also documented. The secondary endpoints—local tumor control (LC) and regional control (RC)—were defined as the time between the end of RT and death or last follow-up (Kaplan–Meier calculation). Overall survival (OS) was analyzed as part of a separate ongoing study. Comparisons between groups were calculated with the Fisher exact test and the log-rank test. The significance threshold was set at 0.2, which is not unusual in exploratory studies [33,34].

## 3. Results

### 3.1. Patients

A total of 78 patients with histologically or cytologically confirmed inoperable stage III NSCLC was included in this analysis. The patient cohort consisted of two groups: 42/78 (53.8%) patients received high-dose CRT (Salzburg) and 36/78 (46.2%) were treated with concomitant CRT (Vienna). After completion of CRT, every patient underwent durvalumab consolidation therapy for one year according to the PACIFIC protocol [1,9]. The mean age in the high-dose (Salzburg) and the SoC group (Vienna) was 64.5 and 62.3 years, respectively. The proportion of individuals younger than 65 years was 47.6% in the Salzburg cohort compared to 57.5% in the SoC arm. At least half of the patients were male (71.4 versus 50% in the SoC group). Patients presented with an ECOG performance status of <2 and approximately 90% or more were current or ex-smokers. In the high-dose group, 38.1% were diagnosed with squamous cell carcinoma and 61.9% with adenocarcinoma, compared to 40.0 and 50.0% in the SoC group. The matching procedure resulted in 29 pairs. From the high-dose cohort, 2/42 (4.8%) had two matches, 1/42 (2.4%) had three and another one (2.4%) had four. It was impossible to find a suitable control for 13/42 (31.0%) high-dose patients. For matching details and patient data, the reader is referred to Appendix A and Table 1, respectively.

### 3.2. Treatment

#### 3.2.1. Chemotherapy

Most patients (97.6%) in the high-dose group received two cycles of platinum-based induction chemotherapy before high-dose radiation treatment, while in the SoC group, the majority underwent four cycles (63.1%) of platinum-based chemotherapy in a concurrent treatment approach (*p* < 0.001). The most commonly used chemotherapy agents in both groups were either carboplatinum/pemetrexed for patients with non-squamous histology or carboplatinum/gemcitabine for squamous histologies. Because of co-morbidities, either carboplatinum or cisplatinum was administered as a single-agent therapy in 8/36 (22.5%) patients.

#### 3.2.2. Radiotherapy

The median total radiation dose to the tumor was 72 Gy (range: 54.0–123.2) in the high-dose group and 59.4 Gy (range: 30.0–70.0) in the SoC group. This amounts to a median biologically effective dose in 2 Gy fractions (*EQD*_2_) of 72 Gy (range: 58.5–121) and 58.4 (range: 32.5–88.4), respectively (Table 2). As expected, these doses differ significantly from each other (*p* < 0.001). Similarly, because the tumor and lymphnode PTVs were significantly smaller in the high-dose group (Mann–Whitney-U test, *p*-values 0.013 and 0.012, respectively) the dose to the lungs was markedly lower: MLD 13.0 Gy versus 16.7 Gy (Mann–Whitney-U test, *p*-value < 0.001) and V20_total lung_ 20.5% vs. 16% (Mann–Whitney-U test, *p*-value < 0.008). In the high-dose cohort, one patient received a second course of irradiation because of an in-field relapse. The first course was administered with 1.8 Gy twice-daily fractions to a total physical dose of 79.2 Gy. In the second course, the patient received 45 Gy in 3 Gy fractions, adding up to a total physical dose of 124.2 Gy, which corresponds to a biological dose of 121 Gy. Similarly, one patient in the SoC cohort had a second course of irradiation with 30 Gy in 3 Gy fractions following primary treatment with 59.4 Gy. Even with these relatively high cumulative doses, no excess toxicity was observed in both patients.

#### 3.2.3. Immunotherapy

The median time intervals from the end of RT to the first course of durvalumab in the high-dose and SoC groups were 18.5 days (range: 4–127) and 22 days (range: 8–114), respectively (Mann–Whitney-U test, *p*-value = 0.841). The median number of 14 (range: 1–26) durvalumab courses administered in the high-dose group differed significantly from the 8 (range: 1–21) courses in the SoC group (Mann–Whitney-U test, *p*-value = 0.031, Table 2).

### 3.3. Toxicity

For the current analysis, emphasis was placed on pulmonary toxicity, including both pneumonitis and pneumonia. The rates of pneumonitis/pneumonia in the high-dose versus SoC groups were 28.6% and in 27.8%, respectively (Table 3, Mann–Whitney-U test, *p*-value = 0.599). The time-to-event analysis revealed that pulmonary toxicity occurred within seven months after the end of radiation treatment (Figure 1, log-rank *p*-value 0.353). The current analysis was designed as a non-inferiority study with a cut-off of +20%. As these percentages in the high-dose and SoC group were 28.6 and 27.8%, respectively, a 95%-one-sided confidence interval for difference could be calculated with 1.64 as the 95%-percentile for normal distribution (0.286 and 0.278 are the toxicity rates in high-dose and SoC group; 1–0.286 and 1–0.278 represent the probability that a patient does not experience pulmonary side effects in either group; 42 and 36 are the patient numbers):$$\left(0.286-0.278\right)+1.64*\;\sqrt{\frac{0.286\;*\;\left(1-0.286\right)}{42}+\frac{0.278\;*\;\left(1-0.278\right)}{36}}=16.8\%$$

As the upper boundary for the confidence interval was 16.8%, which is lower than the cut-off of +20%, high-dose CRT followed by durvalumab was shown to be non-inferior to the SoC with respect to pulmonary toxicity (Appendix A). In general, the rate of side effects was low (Table 3). Apart from the above-mentioned pulmonary toxicity, ICI-mediated hepatitis occurred in 9.5% of patients in the high-dose group without any case in the SoC (Mann-Whitney-U test, *p*-value = 0.089). Durvalumab-related thyreoiditis in high-dose and SoC group was reported in 4.8% and 13.8% (Mann–Whitney-U test, *p*-value = 0.182), respectively. Treatments for immune-mediated toxicities included systemic glucocorticoids (1–4 mg/kg/d prednisone) and endocrine therapy (hypo/hyperthyreoiditis). No treatment-related deaths occurred.

### 3.4. Pulmonary Function Changes after High-Dose Chemoradiotherapy

PFTs were included in the diagnostic work-up of both treatment groups at baseline. PFTs during the course of follow-up were only consistently available in the high-dose cohort. Within a period of six months after the end of RT, 125 PFTs were performed. A total of 39/42 (92.8%) patients completed PFTs, including FEV1 and DLCO, at six months. One had a shorter follow-up and two presented with limited performance status due to disease progression. Of the 39 patients with a follow-up at six months, 24 (61.5%) had a decrease in FEV1. In these patients, the median FEV1 reduction was 3.9 %, with 12 patients showing a reduction of more than 10%. These differences were not statistically significant (Appendix A). As for DLCO, the difference between baseline (=before RT) compared to the end of RT was more pronounced, with a significant decrease in the median DLCO measurement (*p* = 0.002, Appendix A). Interestingly, the decreases in DLCO recovered at six months after radiotherapy without any significant differences to the baseline levels (*p* = 0.853). In 17/39 (43.6%) of patients, the median decrease was 8%, with a decline of more than 10% in seven patients. The relatively moderate changes in PFTs are in line with previous findings, suggesting that high-dose RT is well tolerated with only modest short-term effects on lung function [30].

### 3.5. Local, Regional and Distant Control

The median follow-up was 11.0 months (range: 0.6–40.9) for the 78 patients. Intrathoracic disease control was tendentially better for patients treated in the high-dose group with five (11.9%) versus ten (27.8%) local relapses (Table 4, two-sided Pearson correlation, *p*-value = 0.081) and two (4.8%) versus four (11.1%) isolated regional lymph-node failures (Table 4, two-sided Pearson correlation, *p*-value = 0.184). The comparison between the two groups in the time-to-event analysis revealed a trend towards higher local control (Figure 2, log-rank *p*-value = 0.076) with 91.8% (Salzburg) versus 79.0% (Vienna) at 12 months. As for regional control, the log-rank comparison was non-significant (Figure 3, log-rank *p*-value = 0.313). In 18/78 patients (23.1%) distant metastases were diagnosed with 8/42 (19.0%) cases in the high-dose group and 10/36 (27.8%) individuals in the SoC group (Table 4, two-sided Pearson correlation, *p*-value = 0.261). Again, the time-to-event analysis showed no significant difference (Figure 4, log-rank test, *p*-value = 0.763). Of note, independently from the treatment regimen, the correlation analysis showed a significant impact of local control on regional (Table 4, two-sided Pearson correlation, *p*-value = 0.001) and distant relapse (Table 4, two-sided Pearson correlation, *p*-value = 0.016).

## 4. Discussion

We were able to demonstrate that sequential high-dose chemoradiation followed by durvalumab shows similar pulmonary toxicity rates as SoC (Table 3, Figure 1) and that this regimen yields better intrathoracic disease control (Figure 2 and Figure 3). The results of this study, which is the first to compare high-dose irradiation with SoC, allow us to assume that radiation dose escalation can be safely combined with durvalumab maintenance therapy.

The milestone publication of the PACIFIC data in 2017 revealed a significantly better PFS and OS compared to placebo. PFS at 18-month was 44.2% and 27.0% in the experimental and standard arm, respectively (HR 0.52; 95% CI, 0.42 to 0.65; *p* < 0.001 [1]. The clinical advantage of ICI was corroborated by the 2-year survival data with 66.3% OS for the experimental group versus 55.6% in the control arm (*p* = 0.005). These significant differences were reproducible in follow-up analyses [4,10]. Despite enhanced toxicity of any grade, CRT up to 66 Gy followed by durvalumab was not associated with severe (=grade 3 and 4) excess pulmonary toxicity compared to the placebo arm. The pneumonitis rate grades 3–4 were 3.4 and 2.1% in the treatment and placebo groups, respectively. As for lethal pulmonary side effects, the rate of pneumonitis and other pulmonary events in the durvalumab group was 2%, which was slightly lower than 3.3% in the placebo group [1]. Although the PACIFIC data revealed an absolute increase in PFS at 18 months of 17% [1], the loco-regional failure rate of 36.6% [25] was similar to historical data. Hence it seems that ICI mainly prevents systemic disease progression, whereas the concurring risk of intrathoracic tumor recurrence remains unresolved. This constitutes the rationale for more aggressive local treatment in the context of improved distant control with ICI.

Therefore, Landman et al. investigated the effect of high-dose RT followed by ICI [26] in a cohort of 39 patients. The mean radiation dose was 69.6 Gy, with subsequent durvalumab treatment starting—on average—2.2 months later. The authors reported 15% pneumonitis and—comparable to the PACIFIC trial—one case (3%) of lethal pneumonitis. The probability of pneumonitis was associated with dosimetric parameters such as mean lung dose (MLD). The authors concluded that high radiation doses could be administered without excess toxicity, as long as the dose to the heart is kept low [27,35]. In line with these results, our analysis also demonstrated that pulmonary toxicity is not higher than in the SoC group (Figure 1). Landman et al. reported 21% pulmonary toxicity, including pneumonitis and pneumonia and one case of grade 5 pneumonitis. With approximately 30% in each of the two groups in our analysis, the overall rate of pulmonary events, including radiation and/or ICI-induced pneumonitis and pneumonia, was slightly higher than in the Landman study but lower than the 45% in the PACIFIC study [1]. In contrast, none of the patients in our analysis experienced lethal side effects, although the total radiation dose was higher than 70 Gy and the interval between the end of RT and the start of durvalumab was shorter than in the Landman study [26].

Reasons for the good tolerability of ICI following high-dose CRT in the current analysis may be the sequential treatment modality, which is known to be accompanied by less toxicity than the concurrent treatment approach [5]. In this very same analysis, the 1-year OS rate of 79% [26] was comparable to the PACIFIC trial [1] and RWD [36]. The 21% intrathoracic failures, however, corroborated the notion indicated above, that a more aggressive loco-regional treatment approach may improve disease control in the thorax without excess toxicity. In the high-dose group, 16.7% of patients experienced local or regional failures after 12 months, which was even lower than in the study by Landman. In contrast, the intrathoracic failure rate in the SoC group of the current study was 38.9%, which is in the range of the PACIFIC trial [1,9]. Despite the short follow-up, it is noteworthy that—independently from the treatment regimen—regional (two-sided Pearson correlation coefficient = 0.379, *p*-value = 0.002) and distant control (two-sided Pearson correlation = 0.288, *p*-value = 0.028) seem to be strongly correlated with local control (Table 4). Additionally, the moderate temporary lung function decrease (Appendix A) underlines the fact that high-dose RT is well tolerable paving the way for radical loco-regional treatment combined with ICI.

Faehling et al. analyzed 126 patients from the German early access program, including 32 patients who were PDL1 negative [11]. The median radiation dose was 65 Gy administered with concomitant chemotherapy. Very similar to Landman, the rate of pneumonitis was also 15%, with one case of grade 5 pneumonitis (0.8%). With a 2-year OS of 66% and a median PFS of 20.1 months, these data, similar to other RWD studies [37,38,39,40], corroborate the PACIFIC results. The fact that PD-L1 negative patients had the same oncological outcome challenges the notion that these patients should be excluded from durvalumab therapy based on the disputed EMA decision [41].

An important difference between the above-mentioned studies [1,9,11,26] and our analysis is the fact that CRT in the high-dose group was administered sequentially. The concept of concomitant CRT was established by four prospective randomized phase III trials published between 1999 and 2011 [6,42,43,44] and a meta-analysis [5]. Despite the advantages in clinical outcome in terms of LRC and OS compared to sequential CRT, the concomitant approach is associated with a relative increase in toxicity. The difference with respect to the most prominent side effects after thoracic irradiation, i.e., pneumonitis and esophagitis, is usually a factor of 2, in some cases even 10 [5,43], which is the reason why only 30% of the patients with stage III NSCLC are amenable to concomitant treatment [6]. In contrast, the recently presented PACIFIC-6 data also underline the safe application of durvalumab following sequential CRT in an elderly patient cohort with potentially reduced performance status [40].

The development of advanced radiation technologies, such as intensity-modulated radiotherapy (IMRT) during the last two decades, allows for better sparing of the critical organs at risk, while higher total radiation doses can be applied. Therefore, the follow-up study to the RTOG 9410 [6], the RTOG 0617 published in 2015 [31], tried to evaluate the efficacy of high-dose radiation applied simultaneously with chemotherapy. Unexpectedly, patients in the high-dose radiation arm with 74 Gy in 2 Gy fraction did not only have worse OS but also worse LRC, which left the community with a lot of unanswered questions [45]. One of the reasons for the worse outcome may be the concomitant approach, which may have entailed the increased cardiac toxicities. In contrast, sequentially administered high-dose radiation regimens are tolerated better, and because of the higher total radiation dose that makes up for the delayed start of RT, can be regarded as curative [7,8,27].

The current study has some limitations. Non-inferiority studies usually require high numbers of patients since in small cohorts, statistical differences can be easily overlooked. As this is the first study comparing high-dose irradiation combined with durvalumab in NSCLC stage III, this is a hypothesis-generating analysis to provide the basis for further prospective testing in a larger cohort. Obviously, the smaller PTVs in the high-dose group are advantageous with respect to pulmonary toxicity, which certainly constitutes a bias in the comparison. On the other hand, the high-dose group received significantly more immunotherapy during the observation period (Table 2, *p*-value = 0.031), which harbors a higher risk of pneumonitis. At the present stage, it is unclear in how far the two concurring effects counter-balance each other. Finally, the median follow-up of 11.0 months is too short to adequately assess local, regional and distant control, so that the respective data have to be taken with a grain of salt.

At present, it remains unclear whether the upfront addition of ICI to CRT can be safely administered. Early results from the Keynote 799 trial suggest similar pulmonary toxicitiy rates compared to the SoC with concurrent CRT followed by durvalumab [46], which could pave the way for future treatment strategies. Currently, two studies with a concomitant design including durvalumab are recruiting patients: PACIFIC 2 (ClinicalTrials.gov Identifier: NCT 03519971) and PACIFIC Brasil (ClinicalTrials.gov Identifier: NCT 04230408), with results expected to be published in 2023.

## 5. Conclusions

This hypothesis-generating study suggests that sequential high-dose chemoradiation followed by durvalumab might be similar to SoC in terms of pulmonary toxicity and potentially more effective with respect to intra-thoracic disease control. Larger trials with a prospective design are warranted to validate these results.

## Figures and Tables

**Figure 1 cancers-14-03226-f001:**
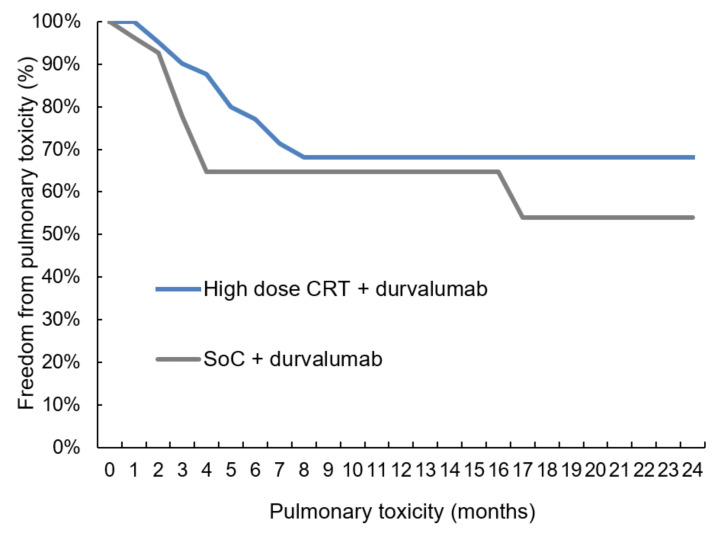
Pulmonary toxicity. The comparison of pulmonary toxicity in a time-to-event analysis revealed no significant difference (log-rank test, *p*-value = 0.353) between high-dose chemoradiation and standard of care (CRT = chemoradiotherapy, SoC = standard of care).

**Figure 2 cancers-14-03226-f002:**
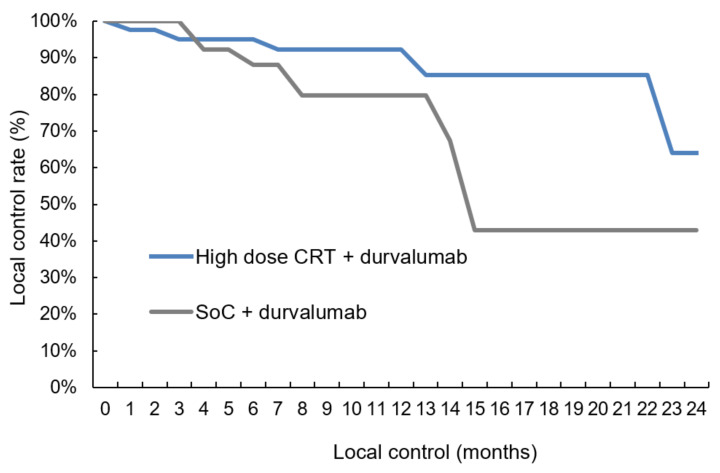
Local control. The comparison showed that local control was tendentially better (log-rank test, *p*-value = 0.076) in the high-dose chemoradiation group than with SoC treatment: The local control rates at 12 months were 91.8% (Salzburg) versus 79.0% (Vienna), respectively (CRT = chemoradiotherapy, SoC = standard of care).

**Figure 3 cancers-14-03226-f003:**
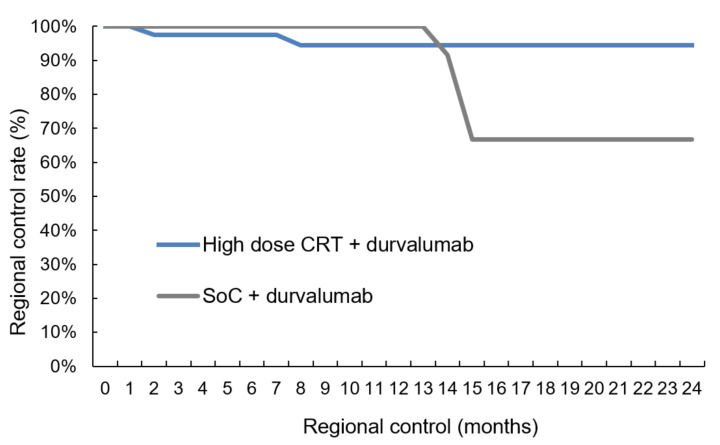
Regional control. The comparison revealed no difference in regional control between high-dose chemoradiation and SoC treatment (log-rank test, *p*-value = 0.313): CRT = chemoradiotherapy, SoC = standard of care.

**Figure 4 cancers-14-03226-f004:**
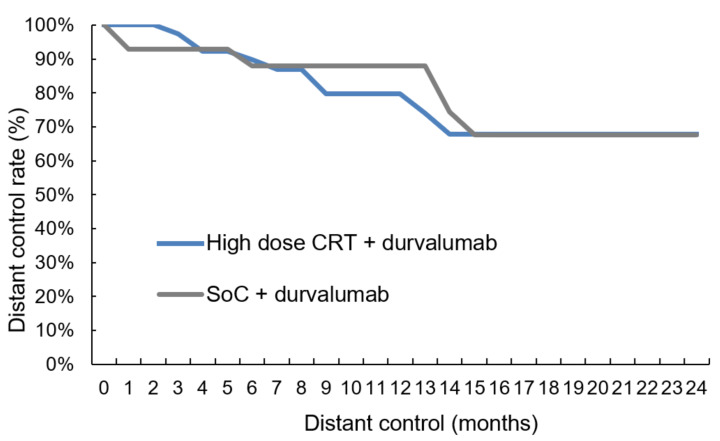
Distant control. The comparison revealed no difference in distant control between high-dose chemoradiation and SoC treatment (log-rank test, *p*-value = 0.763): CRT = chemoradiotherapy, SoC = standard of care.

**Table 1 cancers-14-03226-t001:** Patient characteristics were equally distributed in both groups.

Statistics	High-Dose CRT (Salzburg)	Standard of Care (Vienna)	Statistics
	*n* = 42	*n* = 36	*p*-value
Age			
mean (SD)	64.5 (8.6)	62.8 (8.9)	0.390
≤65 years (%)	47.6	57.5	-
Sex (%)			
female	28.6	50.0	0.411
male	71.4	50.0	-
ECOG (%)			
<2	100	97.2	1
Smoking status (%)			
current, ex-smoker	88.9	97.5	0.278
never	11.1	2.5	-
Histology (%)			
SCC	38.1	40.0	0.799
AC	61.9	50.0	-
NOS	0.0	10.0	-

CRT = chemoradiotherapy, SCC = squamous cell carcinoma, AC = adenocarcinoma, NOS = not otherwise specified, n.a. = not assessed.

**Table 2 cancers-14-03226-t002:** Treatment characteristics.

	Statistics	High-Dose CRT (Salzburg)	Standard of Care (Vienna)	Statistics
		*n* = 42	*n* = 36	*p*-Value
Chemotherapy	Number of cycles (%)			<0.001
	1	2.4	2.5	
	2	97.6	15.0	
	3	0.0	27.5	
	4	0.0	55.0	
	Agents (%)			<0.001
	Carboplatinum/Pemetrexed	59.5	27.5	
	Carboplatinum/Gemcitabine	31.0	0.0	
	Carboplatinum/Taxotere	2.4	0.0	
	Carboplatinum mono	0.0	10.0	
	Carboplatinum/Vinorelbin	0.0	7.5	
	Cisplatinum mono	0.0	12.5	
	Cisplatinum/Pemetrexed	2.4	17.5	
	Cisplatinum/Vinorelbin	0.0	25.0	
	Cisplatinum/Gemcitabine	4.8	0.0	
Radiotherapy	Total dose (Gy)			<0.001
	Median (Min, Max)	72.0 (54.0, 123.2)	59.4 (30.0, 89.4)	
	Biologically effective dose (Gy)			<0.001
	Median (Min, Max)	72.0 (58.5, 121.0)	58.4 (32.5, 88.4)	
	Tumor PTV (mL)			0.013
	Median (Min, Max)	70.5 (9.0, 507)	159.2 (22.7, 939.3)	
	Lymphnode PTV (mL)			0.012
	Median (Min, Max)	100.0 (9.0, 920.0)	268.5 (32.0, 939.3)	
	Mean lung dose (Gy)			<0.001
	Median (Min, Max)	13.0 (6.0, 18.0)	16.7 (6.0, 34.0)	
	V20 total lung (%)			0.008
	Median (Min, Max)	20.5 (6.0, 32.0)	16.0 (5.4, 32.1)	
Immunotherapy	Interval end of RT and start of ICI (days)			0.841
	Median (Min, Max)	18.5 (4.0, 127.0)	22.0 (2.0, 114.0)	
	Cycles (no.)			0.031
	Median (Min, Max)	14 (1, 26)	8 (1, 21)	

CRT = chemoradiotherapy, RT = radiotherapy, ICI = immune checkpoint inhibition, PTV = planning target volume.

**Table 3 cancers-14-03226-t003:** Toxicity: pulmonary toxicity was equally distributed in both groups.

Statistics	High-Dose CRT (Salzburg)	Standard of Care (Vienna)	Statistics
	*n* = 42	*n* = 36	*p*-Value
Pneumonitis or pneumonia (%)	28.6	27.8	0.599
Hepatitis (%)	9.5	0.0	0.089
Thyreoiditis (%)	4.8	13.8	0.182

CRT = chemoradiotherapy.

**Table 4 cancers-14-03226-t004:** This table summarizes the correlation between local and regional as well as distant (= extrathoracic) disease control (two-sided Pearson test).

Clinical Outcome Correlations				
Variable	Regional Control		Distant Control	
Local Control	CC	*p*-value	CC	*p*-value
	0.379	0.001	0.288	0.016

CC = correlation coefficient.

## Data Availability

The data presented in this study are available on request from the corresponding author.

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
