# Peer review of "Durvalumab after Sequential High Dose Chemoradiotherapy versus Standard of Care (SoC) for Stage III NSCLC: A Bi-Centric Retrospective Comparison Focusing on Pulmonary Toxicity"

_cancers, 2022, doi:10.3390/cancers14133226_

Round 1

Reviewer 1 Report

The manuscript by Wass et al. addresses an important medical topic, is nicely written and well understandable. I acknowledge that the authors admit the limitations coming along with the low patient number. The good tolerability of the high dose radiation regime in conjunction with subsequent immunotherapy is the most relevant information of this manuscript, though according to RTOG 617 60 Gy should be standard. In regard to OS, it will be interesting to see how your high dose group performs.  

I have some major and minor comments to be considered:

1. Why is the SoC control group so small? As this represents standard of care it should be quite easy to gather a bigger cohort. Nevertheless, how this control group was employed is not convincing. When using such control in a retrospective design one should perform a matched-pair analysis for better comparison of the two treatment approaches. To argue that there is no statistically significant deviation in baseline parameters is not suitable here as the SoC group is so small and thus less likely to achieve statistically significant differences. Might you provide  a control group of same size as high dose radiation group and perform matched-pair analyses? In reference #31 you give this approach was applied.

2. Introduction: Landman et al. investigated high-dose radiotherapy concomitant with chemotherapy and followed by durvalumab. You should be more precise in delineating that study from your one as you used induction chemo followed by high-dose irradiation. This aspect sould also be addressed in the Discussion.

3. Statistics chapter 2.5: I do not agree with setting the alpha error to 0.2 instead of the usually employed 0.05 threshold. The citations you provide to do do not well apply to your situation. You do not have a prospective phase II trial, in which, under some circumstances, a higher sensitivity at the expense of a lower specificity might be considered. Thus, you should report the associations with local and locoregional relapses as statistical trends without reaching statistical significance and thus, you should be more careful with respective conclusions.

4. Results, 3.2.2 Radiotherapy: Median of 50.4 Gy in the SoC group? In the Methods section on page 3 it is stated that those patients received 60 Gy with fractions of 2 Gy.

5. Results, last chapter of 3.3, page 7: "Durvalumab related thyreoiditis in high dose and SoC group was reported in 4.8% and 26.3%, respectively." There is a conflict with Table 3, in which is stated that 0.0% encountered thyreoiditis in the SoC group.

6. 3.5 Results: Please, provide p-value for comparison of distant metastases between high dose and SoC groups.

Minor issues:

1. Page 3, EQD2: The two radiation regimens in the high dose cohort are not completely identical: EQD2 72,57 vs 71,5 Gy.

2. 2.5 Statistics, first sentence: "designed asa" should be "designed as a".

3. "Pulmonary toxicity, i.e. radiation pneumonitis, durvalumab induced pneumonitis and pneumonia, of any grade was approximately 50% higher in the durvalumab arm than in the control group (46.1% versus 31.2%)." Please, check grammar, not well understandable.

4. Results, chapter 3.1, fifth line: should be "for one year"

5. Results, chapter 3.2.1 chemotherapy: Please, denote the details of "other chemotherapy".

6. Results, chapter 3.4, fourth row: 39/43 patients. Above, there were only 42 patients in the high dose group.

Author Response

The manuscript by Wass et al. addresses an important medical topic, is nicely written and well understandable. I acknowledge that the authors admit the limitations coming along with the low patient number. The good tolerability of the high dose radiation regime in conjunction with subsequent immunotherapy is the most relevant information of this manuscript, though according to RTOG 617 60 Gy should be standard. In regard to OS, it will be interesting to see how your high dose group performs.  

We thank reviewer 1 for his/her in-depth analysis of the paper and the valuable comments, which we tried to address to the best of our knowledge. Please see our point-by-point answers below with insertions in the manuscript text in blue font.

I have some major and minor comments to be considered:

  1. Why is the SoC control group so small? As this represents standard of care it should be quite easy to gather a bigger cohort. Nevertheless, how this control group was employed is not convincing. When using such control in a retrospective design one should perform a matched-pair analysis for better comparison of the two treatment approaches. To argue that there is no statistically significant deviation in baseline parameters is not suitable here as the SoC group is so small and thus less likely to achieve statistically significant differences. Might you provide  a control group of same size as high dose radiation group and perform matched-pair analyses? In reference #31 you give this approach was applied.

Especially this input was very valuable for the paper, as we initially had not considered it. This resulted in a complete overhaul of the whole project. From a pool of 55 patients treated at the Vienna center, 17 individuals were added to the SoC cohort. The matching process is described in detail at the end of the methods and results sections 2.1 Patients and 3.1 Patients. The tables and figures were re-done based on calculations with the new patient numbers (please see below) and the manuscript text was adjusted accordingly.

Abstract

Patients and Methods: Patients with NSCLC stage III received durvalumab after either sequential high dose chemoradiation or concomitant SoC. Chemotherapy consisted of platinum combined with either pemetrexed, taxotere, vinorelbine or gemcitabine depending on histology. The primary endpoint was short term pulmonary toxicity occurring within six months after the end of radiotherapy (RT).

Results: A total of 78 patients was eligible for this analysis. 18F-FDG-PET-CT, cranial MRT and histological/cytological verification was mandatory in the diagnostic work-up. High dose and SoC group included 42/78 (53.8%) and 36/78 (46.2%) patients, respectively, which were matched according to baseline clinical variables. While the interval between end of RT and start of durvalumab was equal in both groups (p = 0.841), more courses were administered in the high dose cohort (p = 0.031). Pulmonary toxicity was similar in both groups (p = 0.599), whereas intrathoracic disease control was better in the high dose group (local control p = 0.081, regional control p = 0.184).

Conclusion: The data of this hypothesis generating study suggest that sequential high dose chemo-radiation followed by durvalumab might be similar to SoC in terms of pulmonary toxicity and potentially more effective with respect to intra-thoracic disease control. Larger trials with a prospective design are warranted to validate these results.

Methods, 2.1 Patients

For every high dose patient we intended to find a suitable match from the SoC group according to the following criteria: age (+/- 10 years), sex, ECOG, smoking status, histology. This approach is similar to a previous study by Johnson [27].

Results, 3.1 Patients

A total of 78 patients with histologically or cytologically confirmed inoperable stage III NSCLC was included in this analysis. The patient cohort consisted of two groups: 42/78 (53.8%) patients received high dose CRT (Salzburg) and 36/78 (46.2%) were treated with concomitant CRT (Vienna). After completion of CRT every patient underwent durvalumab consolidation therapy for year according to the PACIFIC protocol. The mean age in the high dose (Salzburg) and the SoC group (Vienna) was 64.5 and 62.3 years, respectively. The proportion of individuals younger than 65 years was 47.6% in the Salzburg cohort compared to 57.5% in the SoC arm. At least half of the patients were male (71.4% versus 50% in the SoC group). Patients presented with an ECOG performance status of < 2 and approximately 90% or more were current or ex-smokers. In the high dose group 38.1% were diagnosed with squamous cell carcinoma and 61.9% with adenocarcinoma compared to 40.0% and 50.0% in the SoC group. The matching procedure resulted in 32 pairs. While 4/42 (9.5%) individuals in the high dose cohort had more than one control, it was impossible to find a suitable control for 10/42 (23.8%) high dose patients (table 1).

Results, 3.2.1 Chemotherapy (last sentence)

Because of co-morbidities, either carboplatinum or cisplatinum was administered as single agent therapy in 8/36 (22.5%) patients.

Results, 3.2.2 Radiotherapy (third sentence)

Similarly, because the tumor and lymphnode PTVs were significantly smaller in the high dose group (Mann-Whitney-U test, p-values 0.013 and 0.012, respectively) the dose to lungs was markedly lower: MLD 13.0 Gy versus 16.7 Gy (Mann-Whitney-U test, p-value < 0.001) and V20total lung 20.5% versus 16% (Mann-Whitney-U test, p-value < 0.008).

Results, 3.2.3 Immunotherapy

The median time intervals from the end of RT to the first course of durvalumab in the high dose and SoC groups were 18.5 days (range: 4 - 127) and 22 days (range: 8 - 114), respectively (Mann-Whitney-U test, p-value = 0.841). The median number of 14 (range: 1 - 26) durvalumab courses administered in the high dose group differed significantly from the 8 (range: 1 - 21) courses in the SoC group (Mann-Whitney-U test, p-value = 0.031, table 2).

Results, 3.3 Toxicity

The rates of pneumonitis/pneumonia in the high dose versus SoC groups were 28.6% and in 27.8%, respectively (table 3, Mann-Whitney-U test, p-value = 0.599). The time-to-event analysis revealed that pulmonary toxicity occurred within seven months after the end of radiation treatment (figure 1, log-rank p-value 0.353). The current analysis was designed as a non-inferiority study with a cutoff of +20%. As these percentages in the high dose and SoC group were 28.6% and 27.8%, respectively, a 95%-one-sided confidence interval for difference could be calculated with 1.64 as the 95%-percentile for normal distribution (0.286 and 0.278 are the toxicity rates in high dose and SoC group; 1 – 0.286 and 1 – 0.278 represent the probability that a patient does not experience pulmonary side effects in either group; 42 and 36 are the patient numbers):

 = 16.8 %

As the upper boundary for the confidence interval was 16.8%, (…).

In general, the rate of side effects was low (table 3). Apart from the above-mentioned pulmonary toxicity, ICI mediated hepatitis occurred in 9.5% of patients in the high dose group without any case in the SoC (Mann-Whitney-U test, p-value = 0.089). Durvalumab related thyreoiditis in high dose and SoC group was reported in 4.8% and 13.8% (Mann-Whitney-U test, p-value = 0.182), respectively.

Results, 3.5 Local, regional and distant control

The median follow-up was 11.0 months (range: 0.6 – 40.9) for the 78 patients. Intrathoracic disease control was tendentially better for patients treated in the high dose group with five (11.9%) versus ten (27.8%) local relapses (table 4, two-sided Pearson correlation, p-value = 0.081) and two (4.8%) versus four (11.1%) isolated regional lymph-node failures (table 4, two-sided Pearson correlation, p-value = 0.184). The comparison between the two groups in the time-to-event analysis revealed a trend towards higher local control (figure 2, log-rank p-value = 0.076) with 91.8% (Salzburg) versus 79.0% (Vienna) at 12 months. As for regional control, the log-rank comparison was non-significant (figure 3, log-rank p-value = 0.313). In 18/78 patients (23.1%) distant metastases were diagnosed with 8/42 (19.0%) cases in the high dose group and 10/36 (27.8%) individuals in the SoC group (table 4, two-sided Pearson correlation, p-value = 0.261). Again, the time-to-event analysis showed no significant difference (figure 4, log-rank test, p-value = 0.763). Of note, independently from the treatment regimen, the correlation analysis showed a significant impact of local control on regional (table 4, two-sided Pearson correlation, p-value = 0.001) and distant relapse (table 4, two-sided Pearson correlation, p-value = 0.016).

Discussion, third paragraph

Reasons for the good tolerability of ICI following high dose CRT in the current analysis may be the sequential treatment modality, which is known to be accompanied by less toxicity than the current treatment approach (Auperin).

Discussion, fourth paragraph

In the high dose group, 16.7% patients experienced local or regional failures after 12 months, which was even lower than in the study by Landman. In contrast, the intrathoracic failure rate in the SoC group of the current study was 38.9%, which is in the range of the PACIFIC trial. Despite the short follow-up, it is noteworthy that – independently from the treatment regimen – regional (two-sided Pearson correlation, p-value 0.002) and distant control (two-sided Pearson correlation, p-value 0.028) seem to be strongly correlated with local control (table 4).

Discussion, limitations (last but one paragraph, last sentence)

Finally, the median follow-up of 11.0 months is too short to adequately assess local, regional and distant control so that the respective data have to be taken with a grain of salt.

Table 1. Patient characteristics were equally distributed in both groups (CRT = chemoradiotherapy, NOS = not otherwise specified, n.a. = not assessed).

High dose
CRT (Salzburg)

Standard of care (Vienna)

n = 42

n = 36

p-value

Age

mean (SD)

64.5 (8.6)

62.8 (8.9)

0.390

≤ 65 years (%)

47.6

57.5

Sex (%)

female

28.6

50.0

0.411

male

71.4

50.0

ECOG (%)

<2

100

97.2

1

Smoking status (%)

current, ex-smoker

88.9

97.5

0.278

never

11.1

2.5

Histology (%)

SCC

38.1

40.0

0.799

AC

61.9

50.0

NOS

0.0

10.0

Table 2. Treatment characteristics (CRT = chemoradiotherapy, RT = radiotherapy, ICI = immune checkpoint inhibition, PTV = planning target volume, V20total lung = proportion of the total lung volume that receives at least 20 Gy)

High dose
CRT (Salzburg)

Standard of care (Vienna)

n = 42

n = 36

p-value

Chemotherapy

Number of cycles (%)

< 0.001

  1

2.4

2.5

  2

97.6

15.0

  3

0.0

27.5

  4

0.0

55.0

Agents (%)

< 0.001

Carboplatinum/Pemetrexed

59.5

27.5

Carboplatinum/Gemcitabine

31.0

0.0

Carboplatinum/Taxotere

2.4

0.0

Carboplatinum mono

0.0

10.0

Carboplatinum/Vinorelbin

0.0

7.5

Cisplatinum mono

0.0

12.5

Cisplatinum/Pemetrexed

2.4

17.5

Cisplatinum/Vinorelbin

0.0

25.0

Cisplatinum/Gemcitabine

4.8

0.0

Radiotherapy

Total dose (Gy)

< 0.001

Median (Min, Max)

72.0 (54.0 , 123.2)

59.4 (30.0 , 89.4)

Biologically effective dose (Gy)

< 0.001

Median (Min, Max)

72.0 (58.5 , 121.0)

58.4 (32.5 , 88.4)

Tumor PTV (ml)

0.013

Median (Min, Max)

70.5 (9.0, 507)

159.2 (22.7, 939.3)

Lymphnode PTV (ml)

0.012

Median (Min, Max)

100.0 (9.0, 920.0)

268.5 (32.0, 939,3)

Mean lung dose (Gy)

< 0.001

Median (Min, Max)

13.0 (6.0, 18.0)

16.7 (6.0, 34.0)

V20 total lung (%)

0.008

Median (Min, Max)

20.5 (6.0, 32.0)

16.0 (5.4, 32.1)

Immunotherapy

Interval end of RT and start of ICI (days)

0.841

Median (Min, Max)

18.5 (4.0 , 127.0)

22.0 (2.0 , 114.0)

Cycles (no.)

0.031

Median (Min, Max)

14 (1, 26)

8 (1 , 21)

Table 3. Toxicity: the proportion of adverse events – especially pulmonary toxicity – was equally distributed in both groups (CRT = chemoradiotherapy).

High dose
CRT (Salzburg)

Standard of care (Vienna)

n = 42

n = 36

p-value

Pneumonitis or pneumonia (%)

28.6

27.8

0.599

Hepatitis (%)

9.5

0.0

0.089

Thyreoiditis (%)

4.8

13.8

0.182

Table 4. This table summarizes clinical outcome in dependence from treatment regimen (high dose CRT vs. SoC) and the correlation between local and regional as well as distant (= extrathoracic) disease control (two-sided Pearson test): local control was tendentially better in the high dose group and significantly correlated to regional and distant control (n.a. = not assessed).

Clinical outcome correlations

Local control

Regional control

Distant control

High dose CRT vs. SoC

0.081

0.184

0.941

Local control

n.a.

0.001

0.016

Figure 1. Pulmonary toxicity. The comparison of pulmonary toxicity in a time-to-event analysis revealed no significant difference (log-rank test, p-value = 0.353) between high-dose chemo-radiation and standard of care (CRT = chemoradiotherapy, SoC = standard of care).

Figure 2. Local control. The comparison showed that local control was tendentially better (log-rank test, p-value = 0.076) in the high dose group than with SoC treatment. The local control rates at 12 months were 91.8% (Salzburg) versus 79.0% (Vienna), respectively (CRT = chemoradiotherapy, SoC = standard of care).

Figure 3. Regional control. The comparison revealed no difference in regional control between high dose chemo-radiation and SoC treatment (log-rank test, p-value = 0.313): CRT = chemoradiotherapy, SoC = standard of care.

Figure 4. Distant control. The comparison revealed no difference in distant control between high dose chemo-radiation and SoC treatment (log-rank test, p-value = 0.763): CRT = chemoradiotherapy, SoC = standard of care.

  1. Introduction: Landman et al. investigated high-dose radiotherapy concomitant with chemotherapy and followed by durvalumab. You should be more precise in delineating that study from your one as you used induction chemo followed by high-dose irradiation. This aspect sould also be addressed in the Discussion.

We thank reviewer 1 for this important remark and modified the discussion accordingly (3rd paragraph):

Reasons for the good tolerability of ICI following high dose CRT in the current analysis may be the sequential treatment modality, which is known to be accompanied by less toxicity than the concurrent treatment approach (Auperin).

  1. Statistics chapter 2.5: I do not agree with setting the alpha error to 0.2 instead of the usually employed 0.05 threshold. The citations you provide to do do not well apply to your situation. You do not have a prospective phase II trial, in which, under some circumstances, a higher sensitivity at the expense of a lower specificity might be considered. Thus, you should report the associations with local and locoregional relapses as statistical trends without reaching statistical significance and thus, you should be more careful with respective conclusions.

Reviewer 1 is right in as far as the situation of a prospective phase II trial is not entirely comparable with the retrospective setting of the current study. We therefore deleted the last sentence of the statistics section. As requested, we modified the respective passages throughout the manuscript and mitigated our conclusions denoting the differences between groups as trends. In the conclusion, we clearly state that this is a hypothesis generating study, whose results must be validated in independent cohorts. The following parts of the manuscript have been changed.

Abstract, conclusion

The data of this hypothesis generating study suggest that sequential high dose chemo-radiation followed by durvalumab might be similar to SoC in terms of pulmonary toxicity and potentially more effective with respect to intra-thoracic disease control. Larger trials with a prospective design are warranted to validate these results.

Methods, 2.5 Statistics (last sentence deleted)

The significance threshold was set at 0.2, which is not unusual in exploratory studies.

Results, 3.5 Local, regional and distant control

The median follow-up was 11.0 months (range: 0.6 – 40.9) for the 78 patients. Intrathoracic disease control was tendentially better for patients treated in the high dose group with five (11.9%) versus ten (27.8%) local relapses (table 4, two-sided Pearson correlation, p-value = 0.081) and two (4.8%) versus four (11.1%) isolated regional lymph-node failures (table 4, two-sided Pearson correlation, p-value = 0.184). The comparison between the two groups in the time-to-event analysis revealed a trend towards higher local control (figure 2, log-rank p-value = 0.076) with 91.8% (Salzburg) versus 79.0% (Vienna) at 12 months. As for regional control, the log-rank comparison was non-significant (figure 3, log-rank p-value = 0.313). In 18/78 patients (23.1%) distant metastases were diagnosed with 8/42 (19.0%) cases in the high dose group and 10/36 (27.8%) individuals in the SoC group (table 4, two-sided Pearson correlation, p-value = 0.261). Again, the time-to-event analysis showed no significant difference (figure 4, log-rank test, p-value = 0.763). Of note, independently from the treatment regimen, the correlation analysis showed a significant impact of local control on regional (table 4, two-sided Pearson correlation, p-value = 0.001) and distant relapse (table 4, two-sided Pearson correlation, p-value = 0.016).

Conclusion

This hypothesis generating study suggest that sequential high dose chemo-radiation followed by durvalumab might be similar to SoC in terms of pulmonary toxicity and potentially more effective with respect to intra-thoracic disease control. Larger trials with a prospective design are warranted to validate these results.

  1. Results, 3.2.2 Radiotherapy: Median of 50.4 Gy in the SoC group? In the Methods section on page 3 it is stated that those patients received 60 Gy with fractions of 2 Gy.

We apologize for the error. Based on the re-calculation of the data, the numbers have changed so that the first sentence of section 3.2.2 Radiotherapy runs as follows (please also see table 2):

The median total radiation doses to the tumor were 72 Gy (range: 54.0 – 123.2) in the high dose group and 59.4 Gy (range: 30.0 – 70.0) in the SoC group.

  1. Results, last chapter of 3.3, page 7: “Durvalumab related thyreoiditis in high dose and SoC group was reported in 4.8% and 26.3%, respectively.” There is a conflict with Table 3, in which is stated that 0.0% encountered thyreoiditis in the SoC group.

We apologize for this discrepancy. The percentages of thyroiditis were 4.8 (Salzburg) versus 13.8 (Vienna). This was corrected in the manuscript (section 3.3 Toxicity, last paragraph) and in table 3 below.

Durvalumab related thyroiditis in high dose and SoC group was reported in 4.8% and 13.8%, respectively (Mann-Whitney-U test, p-value = 0.182).

Table 3. Toxicity: the proportion of adverse events – especially pulmonary toxicity – was equally distributed in both groups (CRT = chemoradiotherapy).

High dose
CRT (Salzburg)

Standard of care (Vienna)

n = 42

n = 36

p-value

Pneumonitis or pneumonia (%)

28.6

27.8

0.599

Hepatitis (%)

9.5

0.0

0.089

Thyreoiditis (%)

4.8

13.8

0.182

  1. 3.5 Results: Please, provide p-value for comparison of distant metastases between high dose and SoC groups.

We apologize for the negligence. We added a new table (table 4, please see below) summarizing clinical outcome and a new figure (figure 4). Additionally we modified 3.5 Results as follows (please also see above major issues 1 and 3)

Results, 3.5 Local, regional and distant control

The median follow-up was 11.0 months (range: 0.6 – 40.9) for the 78 patients. Intrathoracic disease control was tendentially better for patients treated in the high dose group with five (11.9%) versus ten (27.8%) local relapses (table 4, two-sided Pearson correlation, p-value = 0.081) and two (4.8%) versus four (11.1%) isolated regional lymph-node failures (table 4, two-sided Pearson correlation, p-value = 0.184). The comparison between the two groups in the time-to-event analysis revealed a trend towards higher local control (figure 2, log-rank p-value = 0.076) with 91.8% (Salzburg) versus 79.0% (Vienna) at 12 months. As for regional control, the log-rank comparison was non-significant (figure 3, log-rank p-value = 0.313). In 18/78 patients (23.1%) distant metastases were diagnosed with 8/42 (19.0%) cases in the high dose group and 10/36 (27.8%) individuals in the SoC group (table 4, two-sided Pearson correlation, p-value = 0.261). Again, the time-to-event analysis showed no significant difference (figure 4, log-rank test, p-value = 0.763). Of note, independently from the treatment regimen, the correlation analysis showed a significant impact of local control on regional (table 4, two-sided Pearson correlation, p-value = 0.001) and distant relapse (table 4, two-sided Pearson correlation, p-value = 0.016).

Table 4. This table summarizes clinical outcome in dependence of treatment regimen (high dose CRT vs. SoC) and the correlation between local and regional as well as distant (= extrathoracic) disease control (two-sided Pearson test): n.a. = not assessed.

Clinical outcome correlations

Local control

Regional control

Distant control

High dose CRT vs. SoC

0.081

0.184

0.941

Local control

n.a.

0.001

0.016

Figure 4. Distant control. The comparison revealed no difference in distant control between high dose chemo-radiation and SoC treatment (log-rank test, p-value = 0.763): CRT = chemoradiotherapy, SoC = standard of care.

Minor issues:

  1. Page 3, EQD2: The two radiation regimens in the high dose cohort are not completely identical: EQD2 72,57 vs 71,5 Gy.

We thank reviewer 1 for this valuable point and modified section 2.2 Chemoradiotherapy (3rd paragraph):

Assuming an α/β-value of 10 Gy for tumor tissue, the high dose regimens amount to an EQD2 of 72.6 Gy and 71.5 Gy, respectively, which is is in the range of the dose escalation arm of the RTOG 0617 study with 74 Gy in 2 Gy fractions.

  1. 2.5 Statistics, first sentence: "designed asa" should be "designed as a".

We corrected this sentence accordingly, so that it now runs as follows:

The current bi-center analysis was designed as a non-inferiority study.

  1. "Pulmonary toxicity, i.e. radiation pneumonitis, durvalumab induced pneumonitis and pneumonia, of any grade was approximately 50% higher in the durvalumab arm than in the control group (46.1% versus 31.2%)." Please, check grammar, not well understandable.

Indeed, this sentence does not run fluently, so we changed it as follows and hope that it is clearer now:

This cutoff value was based on the toxicity data of the original PACIFIC study [1]. Any grade pulmonary toxicity, which comprises pneumonitis – either related to radiation or immunotherapy – and pneumonia, was approximately 45% higher in the durvalumab arm than in the placebo group (47% versus 32.5%).

  1. Results, chapter 3.1, fifth line: should be "for oneyear"

Sorry for the negligence, the sentence was corrected:

After completion of CRT every patient underwent durvalumab consolidation therapy for one year according to the PACIFIC protocol.

  1. Results, chapter 3.2.1 chemotherapy: Please, denote the details of "other chemotherapy".

Please refer to the revised table 2 below, which lists all the chemotherapy regimens that were used.

Table 2. Treatment characteristics (CRT = chemoradiotherapy, RT = radiotherapy, ICI = immune checkpoint inhibition, PTV = planning target volume, V20total lung = proportion of the total lung volume that receives at least 20 Gy)

High-dose
CRT (Salzburg)

Standard of care (Vienna)

n = 42

n = 36

p-value

Chemotherapy

Number of cycles (%)

< 0.001

  1

2.4

2.5

  2

97.6

15.0

  3

0.0

27.5

  4

0.0

55.0

Agents (%)

< 0.001

Carboplatinum/Pemetrexed

59.5

27.5

Carboplatinum/Gemcitabine

31.0

0.0

Carboplatinum/Taxotere

2.4

0.0

Carboplatinum mono

0.0

10.0

Carboplatinum/Vinorelbin

0.0

7.5

Cisplatinum mono

0.0

12.5

Cisplatinum/Pemetrexed

2.4

17.5

Cisplatinum/Vinorelbin

0.0

25.0

Cisplatinum/Gemcitabine

4.8

0.0

Radiotherapy

Total dose (Gy)

< 0.001

Median (Min, Max)

72.0 (54.0 , 123.2)

59.4 (30.0 , 89.4)

Biologically effective dose (Gy)

< 0.001

Median (Min, Max)

72.0 (58.5 , 121.0)

58.4 (32.5 , 88.4)

Tumor PTV (ml)

0.013

Median (Min, Max)

70.5 (9.0, 507)

159.2 (22.7, 939.3)

Lymphnode PTV (ml)

0.012

Median (Min, Max)

100.0 (9.0, 920.0)

268.5 (32.0, 939,3)

Mean lung dose (Gy)

< 0.001

Median (Min, Max)

13.0 (6.0, 18.0)

16.7 (6.0, 34.0)

V20 total lung (%)

0.008

Median (Min, Max)

20.5 (6.0, 32.0)

16.0 (5.4, 32.1)

Immunotherapy

Interval end of RT and start of ICI (days)

0.841

Median (Min, Max)

18.5 (4.0 , 127.0)

22.0 (2.0 , 114.0)

Cycles (no.)

0.031

Median (Min, Max)

14 (1, 26)

8 (1 , 21)

  1. Results, chapter 3.4, fourth row: 39/43 patients. Above, there were only 42 patients in the high dose group.

We apologize for the mistake. The high dose group consists of 42 patients.

A total of 39/42 (93%) patients completed PFTs including FEV1 and DLCO at six months.

Reviewer 2 Report

Interesting retrospective study exploring the issue of escalating dose in lung cancer.

Some points should be clarified:

In abstract : 

add definition of "high dose chemoradiotherapy"

Intrathoracic disease control : please add values such a local control 

Patients and methods:

Chemoradiotherapy section :

Please specify the selection of patients: does all patients treated in Salsburg benefit from escalated dose during the inclusion period ? Or why and how those patients were selected to have escalated doses ?

What RT did you deliver ? 3DRT, VMAT ?...

Results:

Toxicity 

Important dosimetric data are mising:

Comparision of dosimetric parameter predicitive of toxicity should have been done such as V5, V20, V30 and mean lung dose... Moreover treatment volume such as PTV CTV GTV should be compared between the 2 centres.

Without these dosimetrical parameters it is difficult to conclude the influence of dose escalation (because of the bias of larger margins in the SOC centre and the influence of low doses on lung toxicity)

3.5 Local, regional and distant control

Local control at 12 months should be clarly mentionned

Author Response

Interesting retrospective study exploring the issue of escalating dose in lung cancer.

We thank reviewer 2 for his/her detailed comments on the manuscript. The issues raised were answered point-by-point with insertions in the manuscript text in blue font.

Some points should be clarified:

  1. In abstract : 

add definition of "high dose chemoradiotherapy"

Intrathoracic disease control : please add values such a local control 

The respective information was added in the abstract

(…) this retrospective bi-center study aims to evaluate pulmonary side effects after sequential high dose chemoradiotherapy beyond 70 Gy compared to SoC.

(…) whereas intrathoracic disease control was better in the high dose group (local control p = 0.081, regional control p = 0.184).

  1. Patients and methods:

Chemoradiotherapy section :

Please specify the selection of patients: does all patients treated in Salsburg benefit from escalated dose during the inclusion period ? Or why and how those patients were selected to have escalated doses ?

What RT did you deliver ? 3DRT, VMAT ?...

Many thanks for pointing out this inaccuracy. In the past 20 years‘ patients with LA-NSCLC have been treated dose escalated regimens at our department (Wurstbauer 2010, 2013; Zehentmayr 2015, Grambozov 2019). They receive two cycles of induction chemotherapy and usually start radiation treatment within one week after the second chemotherapy cycle. In order to make up for the delayed start of radiotherapy they receive higher total doses. We clarified this in the first sentence of section 2.2 Chemoradiotherapy as follows:

Patients at the Salzburg center were traditionally treated with sequential high dose RT after induction chemotherapy (Wurstbauer 2013, Zehentmayr 2015, Grambozov 2019). The patients included in the current analysis received volumetric arc therapy (VMAT).

  1. Results:

Toxicity 

Important dosimetric data are mising:

Comparision of dosimetric parameter predicitive of toxicity should have been done such as V5, V20, V30 and mean lung dose... Moreover treatment volume such as PTV CTV GTV should be compared between the 2 centres.

Without these dosimetrical parameters it is difficult to conclude the influence of dose escalation (because of the bias of larger margins in the SOC centre and the influence of low doses on lung toxicity)

This issue is extremely important therefore we thank reviewer 2 for bringing it up. After adding 17 new patients in the SoC group – as requested by another reviewer – we summarized PTV tumor, PTV lymphnode, V20total lung and MLD in the revised table 2 (please see below). In fact, the comparison between groups revealed that the PTVs as well as the dose to the lungs were significantly smaller in the high dose cohort. For clarity reasons we included only the PTV in the manuscript (table 2).

For the sole purpose of answering the reviewer’s request, we also include a table (please see below: table for reviewer 2), which lists the GTVs. Of note, the planning procedure for the high dose cohort was done on a so-called slow CT including one breath cycle of 4 seconds. This means that the GTVs in the high dose cohort are actually “CTVs”. Hence, a direct comparison with the GTVs of SoC patients would be misleading.

As for the OAR data, the Vienna center could supply us only with reliable data on V20total lung and MLD, so that a comparison between the other parameters mentioned above was not possible. Judging from the data presented in table 2, however, it is reasonable to assume that V5 and V30 would yield similar results.

The discrepancy between the smaller PTVs in the high dose group combined with significantly more cycles of immunotherapy (table 2, p-value = 0.031) certainly constitutes a bias in the comparison. It is unclear, in how far the two concurring effects counter-balance each other. We clearly emphasize this at the end of limitations (Discussion section, last but one paragraph).

Obviously, the smaller PTVs in the high dose group are advantageous with respect to pulmonary toxicity, which certainly constitutes a bias in the comparison. On the other hand, the high dose group received significantly more immunotherapy during the observation period (table 2, p-value = 0.031), which harbors a higher risk of pneumonitis. At the present stage, it is unclear in how far the two concurring effects counter-balance each other.

Table 2. Treatment characteristics (CRT = chemoradiotherapy, RT = radiotherapy, ICI = immune checkpoint inhibition, PTV = planning target volume, V20total lung = proportion of the total lung volume that receives at least 20 Gy)

High dose
CRT (Salzburg)

Standard of care (Vienna)

n = 42

n = 36

p-value

Chemotherapy

Number of cycles (%)

< 0.001

  1

2.4

2.5

  2

97.6

15.0

  3

0.0

27.5

  4

0.0

55.0

Agents (%)

< 0.001

Carboplatinum/Pemetrexed

59.5

27.5

Carboplatinum/Gemcitabine

31.0

0.0

Carboplatinum/Taxotere

2.4

0.0

Carboplatinum mono

0.0

10.0

Carboplatinum/Vinorelbin

0.0

7.5

Cisplatinum mono

0.0

12.5

Cisplatinum/Pemetrexed

2.4

17.5

Cisplatinum/Vinorelbin

0.0

25.0

Cisplatinum/Gemcitabine

4.8

0.0

Radiotherapy

Total dose (Gy)

< 0.001

Median (Min, Max)

72.0 (54.0 , 123.2)

59.4 (30.0 , 89.4)

Biologically effective dose (Gy)

< 0.001

Median (Min, Max)

72.0 (58.5 , 121.0)

58.4 (32.5 , 88.4)

Tumor PTV (ml)

0.013

Median (Min, Max)

70.5 (9.0, 507)

159.2 (22.7, 939.3)

Lymphnode PTV (ml)

0.012

Median (Min, Max)

100.0 (9.0, 920.0)

268.5 (32.0, 939,3)

Mean lung dose (Gy)

< 0.001

Median (Min, Max)

13.0 (6.0, 18.0)

16.7 (6.0, 34.0)

V20 total lung (%)

0.008

Median (Min, Max)

20.5 (6.0, 32.0)

16.0 (5.4, 32.1)

Immunotherapy

Interval end of RT and start of ICI (days)

0.841

Median (Min, Max)

18.5 (4.0 , 127.0)

22.0 (2.0 , 114.0)

Cycles (no.)

0.031

Median (Min, Max)

14 (1, 26)

8 (1 , 21)

Table for reviewer 2. Comparison with Mann-Whitney-U test, p-value 0.009.

Tumor GTV

Salzburg

Vienna

p-value

Median

19

58,9

0.009

Min

0,6

3,5

Max

149

589,3

  1. 5 Local, regional and distant control: Local control at 12 months should be clarly mentioned

As requested by reviewer 2, we added the sentence below in the results section 3.5 stating the 12-months local control rate and the legend to figure 2.

The comparison between the two groups in the time-to-event analysis revealed a trend towards higher local control (figure 2, log-rank p-value = 0.076) with 91.8% (Salzburg) versus 79.0% (Vienna) at 12 months.

Figure 2. Local control. The comparison showed that local control was tendentially better (log-rank test, p-value = 0.076) in the high dose group than with SoC treatment. The local control rates at 12 months were 91.8% (Salzburg) versus 79.0% (Vienna), respectively (CRT = chemoradiotherapy, SoC = standard of care).

Reviewer 3 Report

This is a very relevant study of 2 different regimens for locally advanced NSCLC. Previous studies have not been able to demonstrate an increase in endpoints with an increase in radiation dose from the usual 60-66Gy / 2 gy fractions. The paper is well written.

The main limitation is the lack of randomisation and small size of the study. As such, the methods section should not only mention non-inferiority but also power (number of patients needed). As findings are sometimes referred to as non different when in fact they are just not significant, the authors need to reconsider wording to take the lack of power into account.

A minor point: Authors indicate that radiation fields were larger in the "Vienna" group. This may bias findings so data need to be presented. 

Author Response

This is a very relevant study of 2 different regimens for locally advanced NSCLC. Previous studies have not been able to demonstrate an increase in endpoints with an increase in radiation dose from the usual 60-66Gy / 2 gy fractions. The paper is well written.

  1. The main limitation is the lack of randomisation and small size of the study. As such, the methods section should not only mention non-inferiority but also power (number of patients needed). As findings are sometimes referred to as non different when in fact they are just not significant, the authors need to reconsider wording to take the lack of power into account.
  2. A minor point: Authors indicate that radiation fields were larger in the "Vienna" group. This may bias findings so data need to be presented.

We are grateful for the reviewer’s thorough revision of the manuscript. Please find our point-by-point answers below, insertions in the text are written blue font.

  1. Power calculation

As requested we added a power calculation in section 2.5 Statistics and endpoints, the results of which are described in 3.3 Toxicity (for detailed explanation on assumptions please see below). In concordance with the reviewer’s suggestion, we changed the wording to “non-significant“ instead of “non-different“ throughout the manuscript (e.g. 3.5 Local, regional and distant control), the abstract and the simple summary.

2.5 Statistics and endpoints

Additionally we performed a power calculation based on the following assumptions: α-error 5%, power (1-β) 80%, 47% probability for pulmonary side effects in the control group (S. Antonia, NEJM 2017) and 25% in the high dose group (Wurstbauer 2017, Grambozov 2020). The non-inferiority limit was set at 20% as described above (www.sealedenvelope.com).

  • Toxicity

The rates of pneumonitis/pneumonia in the high dose versus SoC groups were 28.6% and 27.8%, respectively, without a statistically significant difference (table 3, Mann-Whitney-U test, p-value = 0.599). The time-to-event analysis revealed that pulmonary toxicity occurred within seven months after the end of radiation treatment (Figure 1, log-rank test, p-value = 0.353). Again, no difference between groups could be detected.

3.5 Local, regional and distant control

The median follow-up was 11.0 months (range: 0.6 – 40.9) for the 78 patients. Intrathoracic disease control was tendentially better for patients treated in the high dose group with five (11.9%) versus ten (27.8%) local relapses (table 4, two-sided Pearson correlation, p-value = 0.081) and two (4.8%) versus four (11.1%) isolated regional lymph-node failures (table 4, two-sided Pearson correlation, p-value = 0.184). The comparison between the two groups in the time-to-event analysis revealed a trend towards higher local control (figure 2, log-rank p-value = 0.076) with 91.8% (Salzburg) versus 79.0% (Vienna) at 12 months. As for regional control, the log-rank comparison was non-significant (figure 3, log-rank p-value = 0.313). In 18/78 patients (23.1%) distant metastases were diagnosed with 8/42 (19.0%) cases in the high dose group and 10/36 (27.8%) individuals in the SoC group (table 4, two-sided Pearson correlation, p-value = 0.261). Again, the time-to-event analysis showed no significant difference (figure 4, log-rank test, p-value = 0.763). Of note, independently from the treatment regimen, the correlation analysis showed a significant impact of local control on regional (table 4, two-sided Pearson correlation, p-value = 0.001) and distant relapse (table 4, two-sided Pearson correlation, p-value = 0.016).

The power calculation tool can be accessed via: https://www.sealedenvelope.com/power/binary-noninferior/ . This power estimation is based on the following assumptions:

  1. Significance level (α):                 5%
  2. Power (1-β):                 80%
  3. Percentage success in control group: 53%
  4. Percentage success in experimental group: 75%
  5. Non-inferiority limit                 20%
  6. Sample size required per group 22
  7. Total sample size required 44

The assumptions (a) and (b) are conventionally used in medicine.

(c) “Percentage success“ in the current setting means that these patients do NOT have pulmonary toxicity. The number is based on the results of the PACIFIC trial (Antonia, NEJM, table 3 – see screenshot below): in the experimental arm (= patients who received durvalumab, N = 475) pulmonary toxicity occurred in 47% patients (33.9% pneumonitis or radiation pneumonitis, 13.1% pneumonia).

Antonia, S.: NEJM 2017 Nov 16;377(20):1919-1929. doi: 10.1056/NEJMoa1709937

(d) Again, “Percentage success“ in the current setting means that these patients do NOT have pulmonary toxicity. The number is based on previous dose escalation studies by our group (Wurstbauer 2017, Grambozov 2019 – screenshots below). In these analyses the highest rate of acute pulmonary toxicity grade 2 to 5 was 15%. To account for the fact that grade 1 pulmonary toxicity  was not assessed in these analyses, we added 10% to the published rates.

Wurstbauer 2017 Apr;193(4):315-323. doi: 10.1007/s00066-016-1095-4

Grambozov 2020 Feb;11(2):369-378. doi: 10.1111/1759-7714.13276. Epub 2019 Dec 19.

(e) Non-inferiority limit: This number was also based on the original PACIFIC study as described in section 2.5 Statistics and endpoints.

This type of analysis requires definition of a cut-off value below which the experimental arm can be regarded as non-inferior. In the current study, this value was set at +20% meaning that if the high dose group had a maximum of 20% excess pulmonary toxicity compared to SoC (i.e. the confidence interval must not exceed 20%), it could be accepted as non-inferior (supplementary file 1). This cutoff value was based on the toxicity data of the original PACIFIC study. Any grade pulmonary toxicity, which comprises pneumonitis – either related to radiation or immunotherapy – and pneumonia, was approximately 45% higher in the durvalumab arm than in the placebo group (47% versus 32.5%). Based on these data we allowed an excess of 20%, which is less than half of this value, to show non-inferiority.

Antonia, S.: NEJM 2017 Nov 16;377(20):1919-1929. doi: 10.1056/NEJMoa1709937

  1. Minor: larger radiation fields in the Vienna group

This is a very important issue, therefore we thank reviewer 3 for pointing at it. As requested by another reviewer we enlarged the SoC cohort to 78 patients and added the exact PTVs for tumor and lymphnodes in table 2 (please see below). Indeed, the comparison between groups revealed that the PTVs as well as the dose to the lungs were significantly smaller in the high dose cohort.

For the sole purpose of answering the reviewer’s request, we also include a table (table for reviewer 3), which lists the GTVs. Of note, the planning procedure for the high dose cohort was done on a so-called slow CT including one breath cycle of 4 seconds. This means that the GTVs in the high dose cohort are actually “CTVs”. Hence, a direct comparison with the GTVs of SoC patients would be misleading.

The discrepancy between the smaller PTVs in the high dose group combined with significantly more immunotherapy (table 2, p-value = 0.031) certainly constitutes a bias in the comparison. It is unclear, in how far the two concurring effects counter-balance each other. We clearly emphasize this at the end of limitations (Discussion section, last but one paragraph).

Obviously, the smaller PTVs in the high dose group are advantageous with respect to pulmonary toxicity, which certainly constitutes a bias in the comparison. On the other hand, the high dose group received significantly more immunotherapy during the observation period (table 2, p-value = 0.031), which harbors a higher risk of pneumonitis. At the present stage, it is unclear in how far the two concurring effects counter-balance each other.

Table 2. Treatment characteristics (CRT = chemoradiotherapy, RT = radiotherapy, ICI = immune checkpoint inhibition, PTV = planning target volume, V20total lung = proportion of the total lung volume that receives at least 20 Gy)

High dose
CRT (Salzburg)

Standard of care (Vienna)

n = 42

n = 36

p-value

Chemotherapy

Number of cycles (%)

< 0.001

  1

2.4

2.5

  2

97.6

15.0

  3

0.0

27.5

  4

0.0

55.0

Agents (%)

< 0.001

Carboplatinum/Pemetrexed

59.5

27.5

Carboplatinum/Gemcitabine

31.0

0.0

Carboplatinum/Taxotere

2.4

0.0

Carboplatinum mono

0.0

10.0

Carboplatinum/Vinorelbin

0.0

7.5

Cisplatinum mono

0.0

12.5

Cisplatinum/Pemetrexed

2.4

17.5

Cisplatinum/Vinorelbin

0.0

25.0

Cisplatinum/Gemcitabine

4.8

0.0

Radiotherapy

Total dose (Gy)

< 0.001

Median (Min, Max)

72.0 (54.0 , 123.2)

59.4 (30.0 , 89.4)

Biologically effective dose (Gy)

< 0.001

Median (Min, Max)

72.0 (58.5 , 121.0)

58.4 (32.5 , 88.4)

Tumor PTV (ml)

0.013

Median (Min, Max)

70.5 (9.0, 507)

159.2 (22.7, 939.3)

Lymphnode PTV (ml)

0.012

Median (Min, Max)

100.0 (9.0, 920.0)

268.5 (32.0, 939,3)

Mean lung dose (Gy)

< 0.001

Median (Min, Max)

13.0 (6.0, 18.0)

16.7 (6.0, 34.0)

V20 total lung (%)

0.008

Median (Min, Max)

20.5 (6.0, 32.0)

16.0 (5.4, 32.1)

Immunotherapy

Interval end of RT and start of ICI (days)

0.841

Median (Min, Max)

18.5 (4.0 , 127.0)

22.0 (2.0 , 114.0)

Cycles (no.)

0.031

Median (Min, Max)

14 (1, 26)

8 (1 , 21)

Table for reviewer 3. Comparison with Mann-Whitney-U test, p-value 0.009.

Tumor GTV

Salzburg

Vienna

p-value

Median

19

58,9

0.009

Min

0,6

3,5

Max

149

589,3

Reviewer 4 Report

It is an important study to improve regimen for stage III NSCLC therapy. Your article is impressive and give a high impact on cancer therapy.

To assist in understanding and clarifying your paper, I would appreciate it if the following concerns are addressed.

Table 2, page 5 

The letter on the left is hiding by the column. Correct them.

 page 6 in result section 3.3.

Although the calculation formula is described, you should write the definition that can explain the calculation formula in addition to the numbers.

And in the calculation formula, (10316) was (1-0.316)?

page 18 in two graphs

P value you described are surrounded by a square. Correct them.

supplementally data

Please indicate the n number in the figure of supplementary3. What did the significant difference compare to? Describe the method of statistical method. And  add to the box plot and add the raw data in points

Author Response

It is an important study to improve regimen for stage III NSCLC therapy. Your article is impressive and give a high impact on cancer therapy. To assist in understanding and clarifying your paper, I would appreciate it if the following concerns are addressed.

  1. Table 2, page 5 

The letter on the left is hiding by the column. Correct them.

  1. page 6 in result section 3.3.

Although the calculation formula is described, you should write the definition that can explain the calculation formula in addition to the numbers.

And in the calculation formula, (10316) was (1-0.316)?

  1. page 18 in two graphs

P value you described are surrounded by a square. Correct them.

  1. supplementally data

Please indicate the n number in the figure of supplementary3. What did the significant difference compare to? Describe the method of statistical method. And  add to the box plot and add the raw data in points

We thank reviewer 4 for her/his valuable comments, which have definitely led to an improvement of the paper. Please find our point-by-point answers below (insertions in the manuscript are in blue font).

  1. Table 2, page 5

The letter on the left is hiding by the column. Correct them.

Please find the updated table 2 below, the left column should be clearly visible.

Table 2. Treatment characteristics (CRT = chemoradiotherapy, RT = radiotherapy, ICI = immune checkpoint inhibition, PTV = planning target volume)

High-dose
CRT (Salzburg)

Standard of care (Vienna)

n = 42

n = 36

p-value

Chemotherapy

Number of cycles (%)

< 0.001

  1

2.4

2.5

  2

97.6

15.0

  3

0.0

27.5

  4

0.0

55.0

Agents (%)

< 0.001

Carboplatinum/Pemetrexed

59.5

27.5

Carboplatinum/Gemcitabine

31.0

0.0

Carboplatinum/Taxotere

2.4

0.0

Carboplatinum mono

0.0

10.0

Carboplatinum/Vinorelbin

0.0

7.5

Cisplatinum mono

0.0

12.5

Cisplatinum/Pemetrexed

2.4

17.5

Cisplatinum/Vinorelbin

0.0

25.0

Cisplatinum/Gemcitabine

4.8

0.0

Radiotherapy

Total dose (Gy)

< 0.001

Median (Min, Max)

72.0 (54.0 , 123.2)

59.4 (30.0 , 89.4)

Biologically effective dose (Gy)

< 0.001

Median (Min, Max)

72.0 (58.5 , 121.0)

58.4 (32.5 , 88.4)

Tumor PTV (ml)

0.013

Median (Min, Max)

70.5 (9.0, 507)

159.2 (22.7, 939.3)

Lymphnode PTV (ml)

0.012

Median (Min, Max)

100.0 (9.0, 920.0)

268.5 (32.0, 939,3)

Mean lung dose (Gy)

< 0.001

Median (Min, Max)

13.0 (6.0, 18.0)

16.7 (6.0, 34.0)

V20 total lung (%)

0.008

Median (Min, Max)

20.5 (6.0, 32.0)

16.0 (5.4, 32.1)

Immunotherapy

Interval end of RT and start of ICI (days)

0.841

Median (Min, Max)

18.5 (4.0 , 127.0)

22.0 (2.0 , 114.0)

Cycles (no.)

0.031

Median (Min, Max)

14 (1, 26)

8 (1 , 21)

  1. Results section 3.3 Toxicity

Although the calculation formula is described, you should write the definition that can explain the calculation formula in addition to the numbers. And in the calculation formula, (1−0316) was (1-0.316)?

Reviewer 4 is right, the numbers in the formalism need further explanation, which we provided.

As these percentages in the high dose and SoC group were 28.6% and 27.8%, respectively, a 95%-one-sided confidence interval for difference could be calculated with 1.64 as the 95%-percentile for normal distribution (0.286 and 0.278 are the toxicity rates in high dose and SoC group; 1 – 0.286 and 1 – 0.278 represent the probability that a patient does not experience pulmonary side effects in either group; 42 and 36 are the patient numbers):

 = 16.8 %

  1. Page 18

P value you described are surrounded by a square. Correct them.

Indeed, the p-value was surrounded by a square, which was corrected according to the reviewer’s suggestion.

Supplementary file 2

Supplementary file 3

  1. Supplementary data

Please indicate the n number in the figure of supplementary3. What did the significant difference compare to? Describe the method of statistical method. And  add to the box plot and add the raw data in points

Supplementary files 2 and 3 were corrected according to the reviewer’s suggestions (Please also see above, point 3): n was inserted, significant differences refer to baseline, statistical method and raw data points added.

Supplementary file 2. In the high dose cohort moderate changes in FEV1 could be observed at six months after RT compared to baseline (= before RT): n = 39, T-test for paired samples.

Supplementary file 3. In the high dose cohort moderate changes in DLCO could be observed. The statistically significant decrease at the end of RT compared to baseline (= before RT) recovered six months later: n = 39, T-test for paired samples.

Additionally, the parts in section 3.4 Pulmonary function changes after high dose chemoradiotherapy were adjusted as follows.

Of the 39 patients with a follow-up at six months, 24 (61.5%) had a decrease in FEV1 compared to baseline before RT.

(…)

As for DLCO, the difference between baseline (= before RT) compared to end of RT was more pronounced with a significant decrease in the median DLCO measurement (p = 0.002, supplementary file 3).

Round 2

Reviewer 1 Report

Dear authors,

the authors did a great job with substantial improvement of the manuscript.

I have only few items to be considered:

1. Matched-pair analysis. In Methods, 1st chapter, you say that you intended to find a suitable match for each high dose patient. But obviously, you did not achive that according to your statement in the first paragraph of the Results section. Please, harmonize these informations. Moreover, the numbers you provide in the Results part are not completely clear. Once, you state that there were 32 pairs (i.e. 32/42 in the high dose cohort), and that for 10 patients no matching partner could be found. It seems that you had 15 patients of the initial SoC group each matching with one patient of the high dose group and further 17 who were now added. And what about the remaining four patients in the control group? Even more confusing, you state in the Results section that there were 4/42 of the high dose cohort with more than one control? Does this mean that you have actually two or more controls for each of these four high dose patients? Or does it just mean that for four patients of the high dose group there are each more than one patient of the control group so similar that there were theoratically more than one matching option in these four cases? With 32 matching pairs and 36 patients in the control group it looks like that you added four patients who matched with any of the high dose group? But this is not equivalent as to say that 4/42 patients of the high dose cohort had more than one control. Do these four have exactly two matching partners in the control group? Please, specify and be more precise. May you provide a detailed list for all the matched pairs as supplementary material?

2. New Table 4. Besides  that the layout of this table is debatable ("distant control" not set in bold and not encompassed by the horizontal line as a minor issue) this table mixes up the continuous parameter "local control rate" with the dichotomic parameter "high dose CRT vs. SoC"), surely not optimal. And what about the values given? I guess these all reflect p-values and not correlation coefficients. My suggestion is to skip the first row of this table as the respective information is given directly beneath in Fig. 2-4. This might be more appropriate as the Pearson test is not optimal for outcome analysis (i.e. it does not allow censored data). Then, you should instead provide the correlation coefficients in Table 4 along with the already given p-values.

3. Figure 1, y-axis: "Pulmonary toxicity rate"? In the present form it suggests that all patients start with a pulmonary toxicity of 100%. The label should rather indicate freedom from pulmonary toxicity, i.e. rate of patients without pulmonary toxicity.

Author Response

Dear authors,

the authors did a great job with substantial improvement of the manuscript.

I have only few items to be considered:

  1. Matched-pair analysis. In Methods, 1st chapter, you say that you intended to find a suitable match for each high dose patient. But obviously, you did not achive that according to your statement in the first paragraph of the Results section. Please, harmonize these informations. Moreover, the numbers you provide in the Results part are not completely clear. Once, you state that there were 32 pairs (i.e. 32/42 in the high dose cohort), and that for 10 patients no matching partner could be found. It seems that you had 15 patients of the initial SoC group each matching with one patient of the high dose group and further 17 who were now added. And what about the remaining four patients in the control group? Even more confusing, you state in the Results section that there were 4/42 of the high dose cohort with more than one control? Does this mean that you have actually two or more controls for each of these four high dose patients? Or does it just mean that for four patients of the high dose group there are each more than one patient of the control group so similar that there were theoratically more than one matching option in these four cases? With 32 matching pairs and 36 patients in the control group it looks like that you added four patients who matched with any of the high dose group? But this is not equivalent as to say that 4/42 patients of the high dose cohort had more than one control. Do these four have exactly two matching partners in the control group? Please, specify and be more precise. May you provide a detailed list for all the matched pairs as supplementary material?

We thank reviewer 1 for this valuable comment, indeed the respective passages in the manuscript appear unclear to the reader. A list of the matched patients is provided as supplementary file 2 (please see below).

Patient match: high dose patients = red, SoC patients = green

Patient no.

Salzburg = 1, Vienna = 0

Age

Sex: male=1, female=0

0 = ECOG 0 or 1,  1 = ECOG 2 or higher

Smoking status: 1=ex- or current smoker, 0=never smoker

Histology: 1=SCC, 2=AC, 3=NOS

1

1

62,7

0

0

1

2

31

0

52,79

0

0

1

2

11

0

56,35

0

0

1

2

2

1

59,3

0

0

1

1

17

0

56,96

0

0

1

1

14

0

59,83

0

0

1

1

4

0

49,92

0

0

1

1

3

1

61,1

0

0

1

2

5

0

57,01

0

0

1

2

8

0

53,3

0

0

1

2

27

0

55,14

0

0

1

2

33

0

57,96

0

1

1

2

4

1

71,93

0

0

1

2

6

0

65,01

0

0

1

2

5

1

73,98

0

0

1

1

25

0

72,9

0

0

1

2

6

1

69,37

0

0

1

1

13

0

59,35

0

0

1

1

10

0

74,71

0

0

1

1

7

1

72,93

0

0

1

2

2

0

79,47

0

0

1

2

8

1

69,73

0

0

0

1

9

1

69,21

0

0

1

2

3

0

66,51

0

0

1

2

10

1

72,54

0

0

1

2

28

0

74,62

0

0

1

2

11

1

73,68

0

0

0

2

12

0

80,76

0

0

0

3

12

1

70,09

0

0

1

2

18

0

69,1

0

0

1

2

13

1

64,95

1

0

1

1

14

1

63,6

1

0

1

1

7

0

63,17

1

0

1

1

15

1

67,09

1

0

1

2

1

0

75,74

1

0

1

2

16

1

77,22

1

0

0

2

17

1

70,98

1

0

1

2

19

0

62,72

1

0

1

2

18

1

70,56

1

0

1

1

15

0

67,09

1

0

1

1

19

1

32,61

1

0

1

2

9

0

46,54

1

0

1

1

20

1

68,74

1

0

1

1

16

0

62,81

1

0

1

1

21

1

59,06

1

0

1

1

34

0

52,6

1

0

1

1

22

1

66,08

1

0

1

1

20

0

65,91

1

0

1

1

23

1

70,24

1

0

1

2

21

0

69,16

1

0

1

2

24

1

63,16

1

0

1

1

23

0

66,05

1

0

1

3

25

1

58,89

1

0

1

2

26

1

79,54

1

0

1

1

22

0

76,87

1

0

1

1

27

1

52,63

1

0

0

2

28

1

71

1

0

1

1

29

1

73,11

1

0

1

1

29

0

71,09

1

0

1

1

30

1

72,73

1

0

1

2

31

1

56,86

1

0

1

2

24

0

61,71

1

0

1

2

32

1

62,79

1

0

1

1

30

0

66,83

1

0

1

1

33

1

83,41

1

0

1

2

34

1

50,97

1

0

1

2

32

0

46,67

1

0

1

3

35

1

56,2

1

0

1

2

35

0

60,52

1

0

1

2

36

1

63,23

1

0

1

2

26

0

62,92

1

0

1

2

37

1

72,79

1

0

0

2

38

1

68,01

1

0

1

2

39

1

57,54

1

0

1

2

40

1

66,83

1

0

1

2

41

1

59,89

1

0

1

1

36

0

56,03

1

0

1

1

42

1

62,09

1

0

1

2

As for the questions in bold letters, four patients of the high dose group had > 1 match from the SoC group (patients no.1 with two controls; no.2 with three controls; no.3 with four controls; no.6 with two controls), in three patients 1 criterion did not match (patients no. 11,24, 34) and in one patient (patient no. 19) two criteria did not match. For 13 patients (no. 8, 13, 16, 25, 27, 28, 30, 33, 37, 38, 39, 40, 42) we did not find a suitable match. For some odd reason, we erroneously stated in the manuscript that we had 10 patients without a match. This mistake is only explainable by the fact that we made a preliminary match (see table preliminary match below, not included in the manuscript, only for reviewer 1). We did not accept this preliminary match because two male high dose patients (no. 25 and 42) would have had a female SoC match and one high-dose patient (no. 8) would have been closely outside the age range of 10 years. This latter case is a debatable decision, as we accepted another pair (high dose no.19 – SoC no. 9) clearly outside the age range. These latter two, however, were both younger than 50 years of age.

Preliminary match: high dose patients highlighted in red, SoC matches in green – for reviewer 1 only.

Patient no.

Salzburg = 1, Vienna = 0

Age

Gender (male = 1, female = 0)

0 = ECOG 0 or 1,  1 = ECOG 2 or higher

Smoking status: 1 = ex- or current smoker, 0 = never smoker

Histology: 1 = SCC, 2 = AC, 3 = NOS

1

1

62,7

0

0

1

2

31

0

52,79

0

0

1

2

11

0

56,35

0

0

1

2

2

1

59,3

0

0

1

1

17

0

56,96

0

0

1

1

4

0

49,92

0

0

1

1

3

1

61,1

0

0

1

2

8

0

53,3

0

0

1

2

27

0

55,14

0

0

1

2

4

1

71,93

0

0

1

2

6

0

65,01

0

0

1

2

5

1

73,98

0

0

1

1

25

0

72,9

0

0

1

2

6

1

69,37

0

0

1

1

13

0

59,35

0

0

1

1

10

0

74,71

0

0

1

1

7

1

72,93

0

0

1

2

2

0

79,47

0

0

1

2

8

1

69,73

0

0

0

1

14

0

59,83

0

0

1

1

9

1

69,21

0

0

1

2

3

0

66,51

0

0

1

2

10

1

72,54

0

0

1

2

28

0

74,62

0

0

1

2

11

1

73,68

0

0

0

2

12

0

80,76

0

0

0

3

12

1

70,09

0

0

1

2

18

0

69,1

0

0

1

2

13

1

64,95

1

0

1

1

14

1

63,6

1

0

1

1

7

0

63,17

1

0

1

1

15

1

67,09

1

0

1

2

1

0

75,74

1

0

1

2

16

1

77,22

1

0

0

2

17

1

70,98

1

0

1

2

19

0

62,72

1

0

1

2

18

1

70,56

1

0

1

1

15

0

67,09

1

0

1

1

19

1

32,61

1

0

1

2

9

0

46,54

1

0

1

1

20

1

68,74

1

0

1

1

16

0

62,81

1

0

1

1

21

1

59,06

1

0

1

1

34

0

52,6

1

0

1

1

22

1

66,08

1

0

1

1

20

0

65,91

1

0

1

1

23

1

70,24

1

0

1

2

21

0

69,16

1

0

1

2

24

1

63,16

1

0

1

1

23

0

66,05

1

0

1

3

25

1

58,89

1

0

1

2

5

0

57,01

0

0

1

2

26

1

79,54

1

0

1

1

22

0

76,87

1

0

1

1

27

1

52,63

1

0

0

2

28

1

71

1

0

1

1

29

1

73,11

1

0

1

1

29

0

71,09

1

0

1

1

30

1

72,73

1

0

1

2

31

1

56,86

1

0

1

2

24

0

61,71

1

0

1

2

32

1

62,79

1

0

1

1

30

0

66,83

1

0

1

1

33

1

83,41

1

0

1

2

34

1

50,97

1

0

1

2

32

0

46,67

1

0

1

3

35

1

56,2

1

0

1

2

35

0

60,52

1

0

1

2

36

1

63,23

1

0

1

2

26

0

62,92

1

0

1

2

37

1

72,79

1

0

0

2

38

1

68,01

1

0

1

2

39

1

57,54

1

0

1

2

40

1

66,83

1

0

1

2

41

1

59,89

1

0

1

1

36

0

56,03

1

0

1

1

42

1

62,09

1

0

1

2

33

0

57,96

0

1

1

2

Having said all this, we changed the following passages in the manuscript as follows

  1. Methods, 2.1 Patients, last sentence

High dose and SoC patients were matched according to the following criteria: age (+/- 10 years), sex, ECOG, smoking status, histology. This approach is similar to a previous study by Johnson [27].

  1. Results, 3.1 Patients, last two sentences

The matching procedure resulted in 29 pairs. From the high dose cohort, 2/42 (4.8%) had two matches, 1/42 (2.4%) had three and another one (2.4%) had four. It was impossible to find a suitable control for 13/42 (31.0%) high dose patients. For matching details and patient data the reader is referred to supplementary file 2 and table 1, respectively.

  1. New Table 4. Besides  that the layout of this table is debatable ("distant control" not set in bold and not encompassed by the horizontal line as a minor issue) this table mixes up the continuous parameter "local control rate" with the dichotomic parameter "high dose CRT vs. SoC"), surely not optimal. And what about the values given? I guess these all reflect p-values and not correlation coefficients. My suggestion is to skip the first row of this table as the respective information is given directly beneath in Fig. 2-4. This might be more appropriate as the Pearson test is not optimal for outcome analysis (i.e. it does not allow censored data). Then, you should instead provide the correlation coefficients in Table 4 along with the already given p-values.

We agree with reviewer 1 in as far as the mixing up of continuous and discrete variables is not optimal. Therefore – as suggested – we deleted the first line and added the correlation coefficients in the second. Please find the amended table below.

Table 4. This table summarizes the correlation between local and regional as well as distant (= extrathoracic) disease control (two-sided Pearson test): CC = correlation coefficient.

Clinical outcome correlations

Variable

Regional control

Distant control

Local control

CC

p-value

CC

p-value

0.379

0.001

0.288

0.016

  1. Figure 1, y-axis: "Pulmonary toxicity rate"? In the present form it suggests that all patients start with a pulmonary toxicity of 100%. The label should rather indicate freedom from pulmonary toxicity, i.e. rate of patients without pulmonary toxicity.

We apologize for this stupid mistake, please find the corrected figure 1 below.

Figure 1. Pulmonary toxicity. The comparison of pulmonary toxicity in a time-to-event analysis revealed no significant difference (log-rank test, p-value = 0.353) between high-dose chemo-radiation and standard of care (CRT = chemoradiotherapy, SoC = standard of care).
